# On the impact of the parametrization of deep convolutional neural networks on post-training quantization

**Samy Houache**                                    *samy.houache@u-bordeaux.fr*
*Univ. Bordeaux, IMB*
*Thales AVS, France*

**Jean-François Aujol**                             *jean-francois.aujol@math.u-bordeaux.fr*
*Univ. Bordeaux, Bordeaux INP,*
*CNRS, IMB, F-33400, Talence, France*

**Yann Traonmilin**                                 *yann.traonmilin@math.u-bordeaux.fr*
*Univ. Bordeaux, Bordeaux INP,*
*CNRS, IMB, F-33400, Talence, France*

**Reviewed on OpenReview:** *https: // openreview. net/ forum? id= GPsORA7jxD*

## Abstract

This paper introduces novel theoretical approximation bounds for the output of quantized neural networks, with a focus on convolutional neural networks (CNN). By considering layerwise parametrization and focusing on the quantization of weights, we provide bounds that gain several orders of magnitude compared to state-of-the-art results on classical deep convolutional neural networks such as MobileNetV2 or ResNets. These gains are achieved by improving the behaviour of the approximation bounds with respect to the depth parameter, which has the most impact on the approximation error induced by quantization. To complement our theoretical result, we provide a numerical exploration of our bounds on MobileNetV2 and ResNets.

## 1 Introduction

Neural networks have become central to modern machine learning, driving significant advancements across a wide range of applications, including computer vision, natural language processing, and robotics LeCun et al. (2015); Goodfellow et al. (2016). Due to the size of state-of-the-art models, deploying these in resource-constrained environments, such as mobile devices or embedded systems, requires model compression techniques like pruning Han et al. (2016), low-rank approximations Denton et al. (2014) or quantization Gholami et al. (2022). Post-training quantization, in particular, reduces the bit-width of parameters, enabling faster inference and reduced energy consumption without retraining the model. In this paper, we focus on quantized convolutional neural networks because they provide state-of-the-art performances while remaining lightweight, thus, they remain models of choice for embedded applications. Despite the empirical success of quantized models, one concern is the potential performance degradation introduced by quantization. Establishing theoretical bounds on this degradation is therefore crucial, especially for safety-critical applications where robust guarantees are required Forsberg et al. (2020).

For a neural network $R_\theta$ with parameters $\theta$ (i.e. a function parametrized by $\theta$), given an approximation $\theta'$ of $\theta$, we look for bounds of the form

$$\sup_{x \in \Omega} \|R_\theta(x) - R_{\theta'}(x)\|_\infty \leq C\|\theta - \theta'\|_\infty, \tag{1}$$

where $\Omega$ is the domain of the considered inputs of the networks and $C$ is a constant (that must be explicited) depending on the network's architecture. Such bounds quantify the stability of an architecture with respect to parameter perturbations which is essential for understanding the impact of quantization on performance.

Recent works, such as Gonon et al. (2023), provide insightful approximation bounds for quantization. However, these bounds are often pessimistic for practical use and come with strict assumptions. For example, Gonon et al. (2023) imposes a condition that the maximum parameter norm $r$ must be larger than 1, which limits the applicability of their results, particularly when using post-training quantization where the weights are fixed. In practice, for larger networks, we often regularize weights to prevent overfitting Bejani & Ghatee (2021); Santos & Papa (2022). Techniques such as L2 regularization DropConnect Wan et al. (2013) and weight decay Krogh & Hertz (1991) actively encourage the parameters to remain small, potentially making $r$ smaller than 1. Moreover, when removing the impact of weights and input distribution, the constant $C$ in equation 1 exhibits a dependency $O(NL^2)$, where $L$ is the depth of the network and $N$ is the width, making this upper bound of little use for modern deep architectures (which have a large $L$). This opens the following question: is it possible to quantize *deep* neural networks with a practical approximation bound ?

In this work, we provide new theoretical approximation bounds for neural networks, as illustrated in Figure 1. This bound improves several dependencies in the approximation constant $C$ of equation 1 giving a better analysis of the performance of quantized *deep* neural networks in practical cases. We focus on theoretical guarantees for the worst-case quantization error under the *infinity norm* metric, which captures the maximum deviation in network outputs. Our key contributions can be summarized as follows.

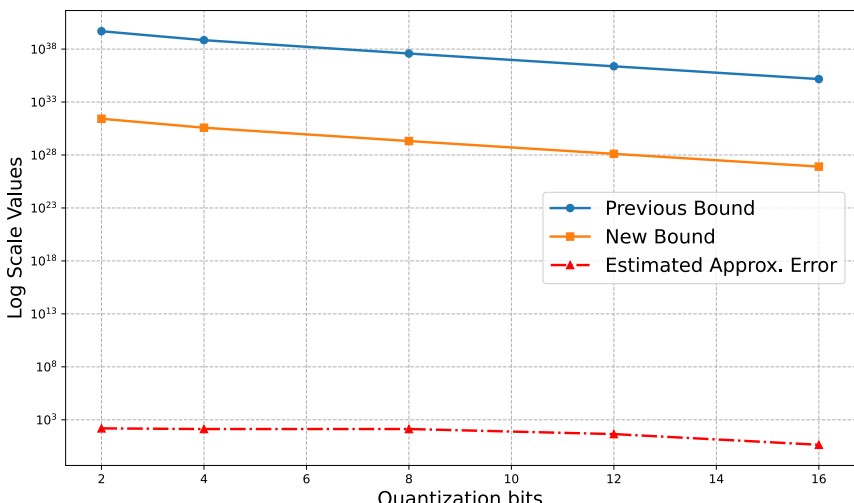

Figure 1: Illustration of the improvement, in log scale, over the previous bound equation 5 on ResNet18 without BatchNorm and without biases, with respect to the number of quantization bits, showing a $10^8$ times tighter error estimation.

**Tighter approximation bounds:** As weights typically require much higher memory usage than biases and many convolutional architectures do not use biases (for the convolutional part), we provide approximation bounds for quantization, when only weights are quantized. In this case, our result enhances the state-of-the-art theorem by Gonon et al. (2023), where we replace their factor of $NL^2$ with the sum $\sum_{\ell=1}^{L} N_{\ell-1}$ where $N_\ell$ is the width of the $\ell$-th layer, which simplifies to $NL$ when the network has a constant width $N$. We further improve this constant for convolutional networks where only the filter sizes and the number of channels of each layer replaces the width in the approximation bound.

**Relaxation of norm constraints:** We weaken the assumptions on the operator norm constraints by considering arbitrary positive values of $r_\ell$, the $l$-th layer operator norm. This generalization makes our result applicable to networks where smaller parameter norms are present due to regularization or sparsity constraints. Also, instead of taking the maximum parameter norm $r$, as in Gonon et al. (2023) and [Corollary F.1]Gonon et al. (2024), our approach replaces it with the less pessimistic expression $r_{mean}$, which (geometrically) averages the parameter norms across layers.

**Practical validation:** We validate our theoretical improvements by applying them to classical pretrained CNN models, demonstrating that our approach is orders of magnitude closer to a practical application compared to previous works.

## 2 Related Works

Approximation bounds have been studied from various perspectives, including results on the approximation capacity of neural networks DeVore et al. (2021); Ding et al. (2019); Csáji et al. (2001); Barron (1993) and the topological properties of the realization map Petersen et al. (2021), particularly focusing on the fact that this realization is Lipschitz continuous with the constant depending on the network architecture Petersen et al. (2021); DeVore et al. (2021).

There also have been analytical works directly quantifying the value of $C$ in Equation equation 1. For example, Neyshabur et al. (2018) studied this constant in the context of a particular case where $\theta'$ is obtained through controlled perturbation (i.e., perturbations that do not significantly modify the norm of the initial weights). They derived, for the $L^2$ norm, a constant that depends on the network depth, the norm of the weights, the data and the perturbation. However, their results do not generalize to any $\theta'$ and are not applicable to arbitrary quantization. Similarly, Berner et al. (2020) expressed the constant in the case of the $L^\infty$ norm but with uniform parameter bounds and, specifically for $d_{\text{out}} = 1$, which cannot be applied to every neural network tasks.

More recently, Gonon et al. (2023) formalized a framework for approximation bounds of ReLU neural networks, providing a general upper-bound for the constant of a neural network in terms of its architecture, weight norms, and other properties. Specifically, their result applies to neural networks defined over general $L^p$-spaces, and under general constraints on the weight parameters. The generalization provided by their upper-bound generalizes prior results, which were often limited to specific cases, such as Neyshabur et al. (2018) for spectrally-normalized networks. By the same authors, in Gonon et al. (2024), there is another approach that generalizes the notion of approximation bounds to Directed Acyclic Graphs (DAGs) using $\ell_1$-path norms. This formulation as graphs allows for more flexibility on the network architecture (pooling, skip connection...). This work provides general bounds with the notable feature of being invariant under parameter rescaling. They improve their previous paper results, relaxing some assumptions, notably the condition $r \geq 1$, but introducing new conditions such as $\theta_i \theta'_i \geq 0$, which is not always satisfied for general distinct parameters $(\theta, \theta')$. We further discuss these bounds in relation to our work in Section 3.

Besides, several works also study such bounds through a statistical point of view, aiming at tighter estimates of the error under typical assumptions (e.g., normality, independence, ...). Such results can be much more informative than deterministic worst-case bounds, but they generally rely on distributional assumptions on the error of weights and/or activations (see, e.g., Sakr et al. (2017)).

A direct application of approximation bounds is quantization. Quantization significantly reduces the storage and computational requirements of deep models Gholami et al. (2022). The impact of quantization on the approximation capabilities of neural networks has been studied in Ding et al. (2019) and Hubara et al. (2018), where the authors provided empirical evidence for maintaining high performance even at low precision. However, formal guarantees are still limited, and existing bounds often assume either uniform quantization or specific activation functions, which limit their applicability. Recent works have introduced strategies for low-bit quantization to retain high predictive accuracy in practical implementations. For instance, Choukroun et al. (2019) explored methods for efficient inference with low-bit quantization, while Courbariaux et al. (2015) demonstrated the effectiveness of binary weight quantization during training, opening a way for training and deploying models with significantly reduced memory and computational demands. Another work that explores methods that optimize weight rounding is AdaRound Nagel et al. (2020) which introduces a data-driven adaptive rounding, which learns rounding offsets to preserve layer outputs post-quantization. This approach can be used to further reduce approximation errors. While our theoretical bounds are designed for general quantization mappings, Adaround can be integrated to tighten the term $\|\theta - \theta'\|_\infty$ in practice. Another important aspect in controlling the accuracy of quantized networks involves understanding singular values in convolutional layers Sedghi et al. (2019) which help inform layer-specific quantization strategies.

## 3 Preliminaries

In this section, we define the key concepts and notations that will be used throughout the paper and we recall reference theoretical approximation bounds from the literature.

**Definition 3.1. Neural network architecture.** The architecture of a neural network is defined by the tuple $(L, \mathbf{N})$, where $L \in \mathbb{N}$ represents the depth of the network, and $\mathbf{N} = (N_0, \ldots, N_L) \in \mathbb{N}^{L+1}$ is a sequence specifying the number of neurons in each layer (the width of each). We call $N_\ell$ the width of the $\ell$-th layer. The width of the network is defined as $N := \max_{\ell=0,\ldots,L} N_\ell$.

**Definition 3.2. Parameters associated with an architecture.** Given an architecture $(L, \mathbf{N})$, parameters associated with this architecture are $\theta = (\tilde{W}_1, \ldots, \tilde{W}_L)$, where $\tilde{W}_\ell \in \mathbb{R}^{N_\ell \times (N_{\ell-1}+1)}$ is the weight matrix for layer $\ell = 1, \ldots, L$ with included bias, i.e. the concatenation of a base weight matrix $W_\ell \in \mathbb{R}^{N_\ell \times N_{\ell-1}}$ with associated bias $b_\ell \in \mathbb{R}^{N_\ell}$, for layer $\ell$. We have that $\theta \in \Theta_{L,\mathbf{N}} := \mathbb{R}^{d(L,\mathbf{N})}$, where the dimension $d(L, \mathbf{N})$ is defined by $d(L, \mathbf{N}) := \sum_{\ell=1}^{L} N_\ell(N_{\ell-1} + 1)$.

**Definition 3.3. Parameter $\ell^\infty$-norm.** For $\theta = (\tilde{W}_1, \ldots, \tilde{W}_L) \in \Theta_{L,\mathbf{N}}$, we define

$$\|\theta\|_\infty := \max_{\ell=1,\ldots,L} \|\tilde{W}_\ell\|_\infty, \qquad \|\tilde{W}_\ell\|_\infty := \max_{i,j} \left|(\tilde{W}_\ell)_{ij}\right|. \tag{2}$$

Since biases are not quantized in our setting and are kept identical in $\theta$ and $\theta'$, the quantity $\|\theta - \theta'\|_\infty$ reduces to the maximum absolute difference over all weight entries.

**Definition 3.4. ReLU Network.** For any vector $x$, we write $\tilde{x} = \begin{pmatrix} x \\ 1 \end{pmatrix}$. Given an architecture $(L, \mathbf{N})$ and parameter vector $\theta = (\tilde{W}_1, \ldots, \tilde{W}_L)$, we associate the function $R_\theta : \mathbb{R}^{N_0+1} \to \mathbb{R}^{N_L}$, which is recursively defined for $\ell = 0, \ldots, L$ as follows:

$$y_0 = x, \ y_\ell = \sigma\left(\tilde{W}_\ell \tilde{y}_{l-1}\right) \text{ and } R_\theta(\tilde{x}) = y_L \tag{3}$$

where $\sigma(x) = \max(0, x)$ is the ReLU activation function.

**Definition 3.5. Domains for parameters and input vectors.** Given an architecture $(L, \mathbf{N})$ and a parameter space $\Theta_{L,\mathbf{N}}$, for any $r \geq 0$, we define the set of admissible parameters:

$$\Theta_{L,\mathbf{N}}(r) := \big\{(\tilde{W}_1, \ldots, \tilde{W}_L) \in \Theta_{L,\mathbf{N}} :$$

$$\|\tilde{W}_\ell\|_{\mathrm{op},\infty} \leq r, \ \ell = 1, \ldots, L\big\}.$$

where $\|\cdot\|_{\mathrm{op},\infty}$ denotes the infinity operator norm, defined as follows, for every matrix $W$ in $\mathbb{R}^{m \times n}$:

$$\|W\|_{\mathrm{op},\infty} := \sup_{x \in \mathbb{R}^n, \ \|x\|_\infty = 1} \|Wx\|_\infty. \tag{4}$$

In this work, we suppose that the input $x$ belongs to the domain $\Omega = [-D, D]^{N_0}$.

We can now give previous approximation bounds in the $\ell^\infty$-norm setting.

**Theorem 3.6** (Previous bound from Gonon et al. (2023)). *For any architecture $(L, \mathbf{N})$, and any $r \geq 1$, denoting $N := \max_{l=0,\ldots,L} N_l$, for any $\theta, \theta' \in \Theta_{L,\mathbf{N}}(r)$, we have :*

$$\sup_{x \in \Omega} \|R_\theta(x) - R_{\theta'}(x)\|_\infty \leq (D+1)NL^2 r^{L-1}\|\theta - \theta'\|_\infty. \tag{5}$$

This shows that the quantization error depends on the architecture's depth $L$, width $N$, and the maximum norm $r$ of weight matrices. As the network depth $L$ increases, the potential error grows exponentially, increasing the sensitivity of deeper networks to small parameter changes.

In Gonon et al. (2024)[Corollary F.1], a new bound is given as a consequence of a general approach using a path-norm metric (see Section 2):

$$\sup_{x \in \Omega} \|R_\theta(x) - R_{\theta'}(x)\|_1 \leq 2\max(D, 1)LN^2 r^{L-1}\|\theta - \theta'\|_\infty \tag{6}$$

The dependency in $L^2$ of bound equation 5 is reduced to a dependency in $L$ at the cost of a dependency in $N^2$ for the $\ell^1$ operator norm. For the sake of clarity in our representations, we have chosen to include only the bound equation 5 in our comparisons. This choice is motivated by the fact that it is generally tighter, particularly because most networks of interest tend to have widths significantly larger than their depths (see Table 1 for an illustration). That is, the condition $2LN^2 \gg NL^2$ often holds, making the Gonon et al. (2023) bound more appropriate for practical comparisons in our $\ell^\infty$ norm setting.

**The convolutional case.** In convolutional networks for image processing, a convolutional layer can be represented as a matrix multiplication. Consider an input image $x$ of dimensions $n \times m$ and a convolutional filter $h$ of dimensions $p \times p$. The input image $x$ can be flattened into a column vector $\mathbf{x}$ of size $nm \times 1$: $\mathbf{x} = \text{vec}(X)$ where $\text{vec}(\cdot)$ denotes the vectorization operation. The convolution of the input image $x$ by the filter $h$ of size $p \times p$ followed by a subsampling/upsampling can be expressed as a matrix product $\mathbf{y} = \mathcal{H}\mathbf{x}$ where $\mathbf{y}$ of dimensions $n_\ell \times m_\ell$ is the vectorization of the output image and $\mathcal{H}$ is a matrix representing the 2d convolution by $h$ (hence each row contains at most $p^2$ coefficients). The weight matrix associated with a given convolutional layer is a collection of $c_\ell$ convolutions by several filters $h_\ell$, followed by a subsampling or an upsampling. Hence the width of the $\ell$-th layer is $c_{\ell-1} \times n_{\ell-1} \times m_{\ell-1}$.

## 4  Theoretical results

In this section, we present our main theoretical results, which extend and improve upon the existing bounds for quantized ReLU networks by relaxing the constraints on the network parameters and considering a quantization of the weight matrices only (and not biases). Indeed, in practice, convolutional neural networks such as MobileNetV2 or Resnets without biases are used successfully for vision tasks. If we consider more general networks with biases (and a constant width for the sake of discussion), the size of the bias vector is $NL$ compared to the size of weight matrix $N^2L$. If the width $N$ of the network is large compared to the objective in terms of memory requirement, e.g. $N >> 8$ for a $8\times$ memory reduction from 64 bits to 8 bits, then quantifying biases will add little gain in memory compared to the gain resulting from the quantization of weights (this is seen in practice when the biases often remain in full precision or are only lightly quantized Finkelstein et al. (2019)).

Although we introduce the setting with ReLU for clarity, our theoretical analysis does not rely on piece-wise linearity. All results in this section hold for any activation function $\sigma$ that is 1-Lipschitz and satisfies $\sigma(0) = 0$ (e.g., ReLU, leaky ReLU, tanh, softsign), as also illustrated in Appendix I with tanh.

We first give a general approximation bound and then specify to the convolutional case.

**Theorem 4.1** (General approximation bound). *For any architecture $(L, \mathbf{N})$, define the parameters $\theta = (\tilde{W}_1, \ldots, \tilde{W}_L)$ and $\theta' = (\tilde{W}'_1, \ldots, \tilde{W}'_L)$, where $\tilde{W}_\ell$ and $\tilde{W}'_\ell$ are weight matrices with included bias. Assume that the two networks have same biases. Assume besides that $\forall \ell = 1, \ldots, L$:*

$$\|\tilde{W}_\ell\|_{\text{op},\infty} \le r_\ell \quad and \quad \|\tilde{W}'_\ell\|_{\text{op},\infty} \le r_\ell. \tag{7}$$

*Then:*

$$\sup_{x \in \Omega} \|R_\theta(\tilde{x}) - R_{\theta'}(\tilde{x})\|_\infty \le \max(D, 1) \sum_{\ell=1}^{L} N_{\ell-1} \times r_{mean}^{L-1} \|\theta - \theta'\|_\infty, \tag{8}$$

*where we define the mean norm parameter*

$$r_{mean} := \sqrt[L-1]{\max_{l=1,\ldots,L} \max_{i=1,\ldots,l-1} \prod_{\substack{j=i \\ j \ne l}}^{L} r_j}. \tag{9}$$

Note that the maximum norm parameter $r$ in equation 5, from Gonon et al. (2023), is replaced by the term $r_{mean}$. This geometric mean-like term, which considers the largest of partial products of layer-wise norm parameters, allows for a better adjustment of the bound to the variability of the norms. This variability

can be significant, particularly between $r_{\max}$ and the other $r_l$ values. Specifically, in the least favorable case where $r_1 = r_2 = \cdots = r_L$, the largest product simplifies to $r^{L-1}$. Conversely, the most favorable scenario would involve a distribution of $r_l$ with high variance, particularly where $\max(r_l) \gg 1$ and all other $r_l \leq 1$. In such cases, $r_{mean}^{L-1}$ would be much smaller than $r^{L-1}$. We provide a more in depth discussion about this, in Appendix G, with simulated examples in Figure 10 .

*Remark* 4.2 (Improved factor $\sum_{l=1}^{L} N_{l-1}$). Our result introduces an improved factor, which takes into account the sum of the layer widths. If the architecture has uniform width across all layers, i.e., all layers have width $N$, this factor becomes $N \times L$, which is smaller than $N \times L^2$ of Equation equation 5 .

*Remark* 4.3 (Weakened condition for the domain of parameters $r_\ell$). In this bound, the constants $r_\ell$ are allowed to be arbitrary positive numbers. This weakens the condition $r \geq 1$ of previous bound, making the result more general and applicable to a wider range of network architectures.

**Theorem 4.4** (Approximation bound for CNN). *With the same settings as in Theorem 4.1 and for a purely convolutional network without biases, where each layer applies $c_l$ filters of size $p_l \times p_l$, we have:*

$$\sup_{x \in \Omega} \|R_\theta(x) - R_{\theta'}(x)\|_\infty \leq D \times \sum_{l=1}^{L} p_l^2 c_{l-1} \times r_{conv}^{L-1} \|\theta - \theta'\|_\infty \tag{10}$$

*where we define*

$$r_{conv} := \sqrt[L-1]{\max_{l=1,\ldots,L} \prod_{\substack{k=1 \\ k \neq l}}^{L} r_k^{conv}} \tag{11}$$

*with $r_k^{conv}$ a bound on the norm of the convolutional matrix of layer $k$ without bias (i.e. $r_k^{conv} \geq \|\mathcal{H}_k\|_{op,\infty}$).*

With this formulation, the term $\sum_{\ell=1}^{L} p_\ell^2 c_{\ell-1}$ accounts for the sparse structure of the convolution matrices $\mathcal{H}_\ell$ rather than simply using the number of row elements $N_{l-1} = m_{l-1}n_{l-1}c_{l-1}$ ($m_{l-1}n_{l-1}$ being the size of each channel at each layer), which can be much larger, see Table 1). Then notice that we use $r_{conv}$ in this bound, which corresponds to the larger product omitting one term and starting from the first layer, so $r_{conv}$ is smaller than $r_{mean}$ by construction. Finally, we also have an intermediate result between Theorem 4.1 and Theorem 4.4, which corresponds to the case of MLP without bias, given in Appendix A (Theorem A.4).

**Corollary 4.5** (Approximation bound for a stack of ResNet-18 residual blocks). *Consider a sequence of $B$ residual blocks defined in Lemma B.1, without batch normalization (as in our experimental setting). We assume that the activation $\sigma$ is 1-Lipschitz and satisfies $\sigma(0) = 0$.*

*Let $\theta$ and $\theta'$ denote the collections of all weight matrices*

$$\theta = (W_1^{(1)}, W_2^{(1)}, \ldots, W_1^{(B)}, W_2^{(B)}), \qquad \theta' = (W_1^{(1)\prime}, W_2^{(1)\prime}, \ldots, W_1^{(B)\prime}, W_2^{(B)\prime}). \tag{12}$$

*So we have $L = 2B$ layers, where the $\ell$-th layer corresponds to either $W_1^{(b)}$ or $W_2^{(b)}$ depending on b. Consider additionally shortcut matrices $W_s^{(b)}$ defining skip connections for each block.*

*Define, for $\ell = 1, \ldots, 2B$,*

$$r_\ell := \begin{cases} \max(1, \|W_1^{(b)}\|_{op,\infty}), & \text{if } \ell = 2b-1, \\ \|W_2^{(b)}\|_{op,\infty} + \|W_s^{(b)}\|_{op,\infty}, & \text{if } \ell = 2b. \end{cases} \tag{13}$$

*Define*

$$r_{\text{mean}}^{\text{ResNet}} := \sqrt[2B-1]{\max_{l=1,\ldots,2B} \max_{i=1,\ldots,l-1} \prod_{\substack{j=i \\ j \neq l}}^{2B} r_j}. \tag{14}$$

*Thus, we get the following bound over any input domain $\Omega \subset [-D, D]^{N_0}$:*

$$\sup_{x \in \Omega} \|R_\theta(\tilde{x}) - R_{\theta'}(\tilde{x})\|_\infty \leq \max(D, 1)\,(r_{\text{mean}}^{\text{ResNet}})^{2B-1} \sum_{\ell=1}^{2B} p_\ell^2 c_{\ell-1}\,\|\theta - \theta'\|_\infty. \tag{15}$$

The proof can be obtained by applying Theorem 4.4 to this architecture. A similar corollary can be established for the ResNet-50 architecture.

## 5 Experiments

In this section, we describe the methodology and setup used to validate our theoretical findings, followed by the results of our experiments, that involve *pretrained* architectures. We evaluate the performance of larger models such as **ResNet18**, **ResNet50**, and **MobileNetV2**, all pretrained on the ImageNet dataset Deng et al. (2009). These models use deep convolutional architectures without biases (except the last fully connected layer) that are frequently used in real-world applications. Note that a particular care must be taken to manage skip connections (see Lemma B.1 and Lemma B.2 in the Annex) in the calculations. For experiments with these networks, we removed the BatchNormalization of the networks to match the conditions of Theorem 4.4. By doing this, we still obtained similar accuracies to models with BatchNorm (increasing the number of epoch). More precisely, the degradation of accuracy remains small across all architectures (arround 1% for ResNet18 and MobileNetV2, and below 2% for ResNet50). Moreover Batch Normalization is not guaranteed to be 1-Lipschitz in general, since its scale and shift parameters are learned during training. One could assume the $\ell$-th BN layer is $\beta_\ell$-Lipschitz with $\beta_\ell \geq 1$, which would introduce a factor $\prod_{\ell=1}^{L} \beta_\ell$ in the final bound. So this would just complicate the analysis and loosen our bound which is already conservative.

The goal of these experiments is to empirically compare the bounds given by our new theoretical results against those derived from the work of Gonon et al. (2023). The pretrained models are also used to evaluate the effects of post-training quantization on model performance. We conduct experiments on the Tiny ImageNet dataset, which contains a total of 110,000 images resized to $64 \times 64$ pixels. It includes 100,000 training images, 10,000 test images, and covers 200 distinct classes selected from the original ImageNet. Other experiments on the MNIST and CIFAR-10 datasets leading to similar conclusion are presented in the Annex E.

### 5.1 Analysis of weight distribution across layers

The results of Figure 2 highlight the key advantage of our theoretical bound equation 10 compared to prior works. In Gonon et al. (2023), the bound depends on the maximum operator norm $r$, which in these examples, is significantly larger than the "geometric mean" layer-wise term $r_{conv}$. For example, in ResNet50, the maximum $r$ is approximately 3 times larger than $r_{conv}$, and in MobileNetV2, it is more than 11 times larger. Specifically for MobileNetV2 we can see that the weight norm distribution looks like the exponential distribution of Figure 10, that is a favorable case to have a tighter bound. Lots of values are small and the maximum value $r$ is more than 100, while $r_{conv}$ is 9. This shows better consideration of network weight norm distribution. In practice, techniques such as Cross-Layer Equalization (CLE) Nagel et al. (2019) can be used as a preprocessing step to homogenize weight distributions across layers for the quantized network. By doing this the value of $r_{conv}$ decreases and leads to tighter bounds in our framework. For example in Figure 11 (in Appendix H), when applying CLE to a 4-bit quantized ResNet50 using the AIMET library Siddegowda et al. (2022), we observed that $r_{conv}$ went from 6 to 4 after CLE. This shows how our theoretical results can explain the benefits of such transformations.

### 5.2 Quantization

In our experiments, we consider three types of **post-training quantization**. We define *uniform* quantization as a function $Q^{unif} : \Theta_{L,\mathbf{N}} \to \Theta_{L,\mathbf{N}}$, where $Q(\theta)$ represents the quantized parameter. Specifically, we use the following definition of uniform quantization in our experiments:

$$Q^{unif}(\theta) = Q_\eta^{unif}(\theta) = \left\lfloor \frac{\theta}{\eta} \right\rfloor \eta, \tag{16}$$

where $\eta > 0$ is the quantization step size. For each layer, the parameter $\eta$ is determined by the formula:

$$\eta = \frac{W_{\max}}{2^n - 1},$$

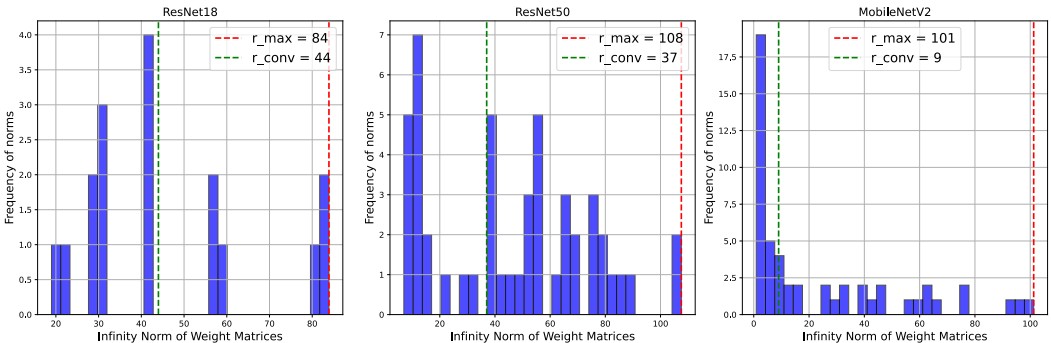

Figure 2: Comparison between the maximum geometric mean term $r_{\text{conv}}$ (green) used in equation 10 and the maximum weight norm $r$ (red) used in equation 5, for ResNet18, ResNet50 and MobileNetV2, without BatchNorm, showing a smaller value of $r_{conv}$ for all models.

where $W_{\max}$ is the maximum absolute value of the weight values, and $n$ is the bit width.

To highlight the role of the term $\|\theta - Q(\theta)\|_\infty$ in the quantization error, as an alternative approach to uniform quantization, we define $Q^{round}$ using a rounding function instead of floor truncation. This form of quantization is defined as:

$$Q^{round}(\theta) = Q^{round}_\eta(\theta) = \text{round}\left(\frac{\theta}{\eta}\right)\eta, \tag{17}$$

where the function round rounds $\frac{\theta}{\eta}$ to the nearest integer before scaling back by $\eta$. This approach may reduce quantization error in cases where uniform quantization leads to excessive loss of information.

In the same way, we also consider an adaptive quantization approach, which is a simplified AdaRound scheme. AdaRound optimizes a per-parameter rounding offset via a differentiable relaxation of the binary rounding decision. More precisely, for a given parameter $\theta$, our adaptive quantization is defined as:

$$Q^{adaround}(\theta) = \left(\left\lfloor\frac{\theta}{\eta}\right\rfloor + \sigma(\alpha)\right)\eta, \tag{18}$$

where $\alpha$ is a learnable offset, and $\sigma(\cdot)$ denotes the sigmoid function. The offset $\alpha$ is optimized over a calibration dataset by minimizing the mean squared error between the FP32 output and the quantized one, with an additional regularization term that encourages $\sigma(\alpha)$ to converge to 0.5 (i.e., standard rounding behavior). This approach better adjusts the quantization with the distribution of the weights.

### 5.3 Experimental results on MobileNetV2 and Resnets

We analyze key parameters involved in our bounds and those from Gonon et al. (2023) in Table 1. The depth $L$ of the considered architectures varies between $L = 18$ for ResNet18 and $L = 53$ for MobileNetV2. This variation in depth is important, as the approximation bound is exponentially dependent on $L$. An important difference is observed in the width parameter $N$, where the values obtained using the formulation of Theorem 4.4 bound are orders of magnitude smaller than those from the previous bound. For instance, for MobileNetV2, the previous bound is calculated with a width of $1.2 \times 10^6$, whereas the new bound reduces this to 8641. Similarly, for ResNet50, the width decreases from $8 \times 10^5$ to 4609. This significant decrease reflects the tighter characterization provided by the new bound, which avoids overly conservative estimations of $N$. Another critical parameter is the maximum norm parameter, which is significantly smaller under the new bound, particularly for MobileNetV2 ($r \approx 101$) while $r_{mean}$ only equals to 9. The reduced values of norm parameters reduce the exponential dependency on depth, which is the main pessimistic factor in the bound. These differences between the two bounds is reflected in the ratio value. Even for a shallow network like ResNet18, we notice in Table 1 that our bound is $10^8$ times tighter and this observation becomes even more relevant for deeper and wider networks such as MobileNetV2, where the ratio reaches $10^{56}$.

Table 1: Comparison of parameters between our bound equation 10 and the state-of-the-art Gonon et al. (2023) bound on pre-trained models. The comparison is also expressed in terms of a ratio of the two bounds, where the values of the bounds in this ratio are computed exclusively for the convolutional part of the network.

| | | Previous Width | Previous Norm Param | New Width | New Norm Param | Ratio |
| Model | Depth ($L$) | $m_{l-1}n_{l-1}c_{l-1}$ | $r$ | $p_l^2 c_{l-1}$ | $r_{conv}$ | Previous Bound (2023) / New bound |
| MobileNetV2 | 53 | $1.2 \times 10^6$ | $\approx 101$ | 8641 | $\approx 9$ | $\approx 10^{56}$ |
| ResNet18 | 18 | $8 \times 10^5$ | $\approx 84$ | 4609 | $\approx 44$ | $\approx 10^8$ |
| ResNet50 | 50 | $8 \times 10^5$ | $\approx 108$ | 4609 | $\approx 37$ | $\approx 10^{27}$ |

In Figure 1 (in the introduction), we compare the approximation error bound and the estimated approximation error for the Resnet18 architecture. The results were obtained by removing the final fully connected layer to keep only the convolutional part of the network. Then we use $r_{conv}$ to calculate the new bound since the network has no biases for the convolutional part. The figure demonstrates an error of approximately $10^{40}$, whereas the new bound is only $10^{32}$. This highlights a significant improvement, even for a shallow network (with $L = 18$). In Figure 3a, we perform the same experiment for the MobileNetV2 architecture. The effect of weight norm distribution is even more apparent. The previous bound reaches a value of $10^{109}$, whereas the new bound is reduced to $10^{53}$. For both figures, we can observe that the shape of the bound is accurate as it follows the error approximation trend, up to a constant factor. Nevertheless, the constant is still large even for the new bound, since the output error is at most around $10^3$.

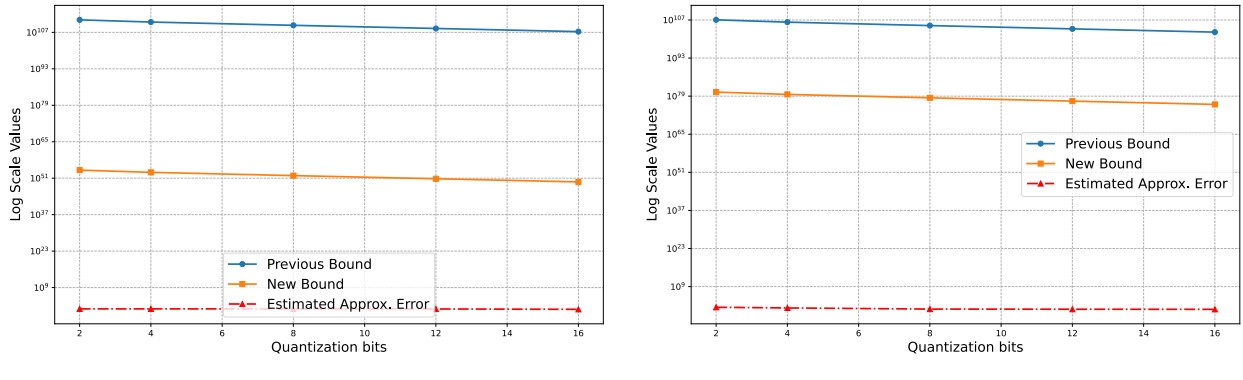

(a) MobileNetV2 without BatchNorm and biases.

(b) ResNet50 without BatchNorm and biases.

Figure 3: Comparison in log scale between our bound equation 10 and the previous bound equation 5, on the convolutional part of (a) MobileNetV2 and (b) ResNet50, with respect to the number of bits. Our bound is approximately $10^{56}$ times tighter for MobileNetV2 and $10^{27}$ times tighter for ResNet50.

We can make similar observations from Figure 3b because the previous bound has the same order of magnitude. However, even though MobileNetV2 and ResNet50 have similar depths, our bound has adapted much better to the specific distribution of weight norms in MobileNetV2. The bound value for MobileNetV2 is significantly lower than the ResNet50 one, being arround $10^{26}$ times smaller. This is mainly due to the favorable distribution of weight norms in MobileNetV2, which favors $r_{conv}$ to remain small. Although our new bound significantly improves the previous state of the art bound (e.g., Gonon et al. (2023), [Corollary F.1]Gonon et al. (2024)), the new bound is still orders of magnitude far from what is observed in real outputs of networks (Figures 1,3a and 3b). One reason is that such bounds are derived under theoretical worst-case scenarios, which rarely occur in practice. Another reason is the generality of the theorem itself, which relies on inequalities such as

$$\|Ax\|_\infty \le \|A\|_{\mathrm{op},\infty} \|x\|_\infty, \tag{19}$$

which can become conservative when the intermediate activations are not aligned with the worst-case directions of the matrices. This type of misalignment helps explain the gap between worst-case guarantees and typical behavior observed in practice.

In Figure 4 we analyze the impact of post-training quantization on the accuracy of MobileNetsV2 and ResNets, across Tiny Imagenet. We notice that the Adaround method gives similar performances to the round method, except for ResNet18 where the behavior for extreme quantization ($\leq 4$ bits) is better. Indeed for ResNet18 the accuracy with 4 bit quantization is over 30% with the AdaRound method, while it is below 10% accuracy with floor and round methods. Finally, it appears that MobileNetV2 better support quantization with floor method than ResNets. This is probably due to the specific architecture of MobileNetV2 that uses residual bottlenecks and Relu6 activation function ($ReLu6(x) := \min(\max(0, x), 6)$ which is known to better support quantization Sandler et al. (2018). Furthermore, the dataset influences precision as well, that is why we conducted similar experiments across MNIST LeCun (1998) and CIFAR-10 Krizhevsky et al. (2009) in which we can also observe the behavior reflected by the form of the bound: depth significantly influences quantization errors (see Figure 9 in the Appendix E).

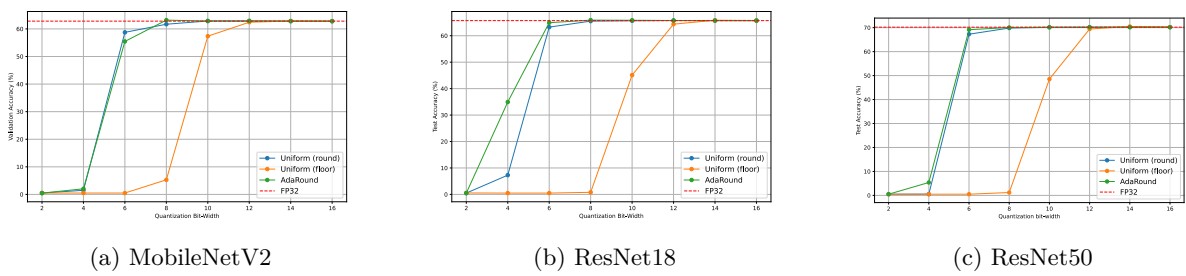

(a) MobileNetV2       (b) ResNet18       (c) ResNet50

Figure 4: Graphs illustrating the effect of quantization on performance on Tiny ImageNet for three quantization functions (round, uniform and Adaround). The results highlight how quantization reduces memory requirements while maintaining or approaching the base model's accuracy. The amount of quantization needed to reach the base precision depends on the quantization function used.

### 5.4 Effect of depth in the quantization of MLP without biases

In Figure 5a, we investigate the improvement in accuracy of our bound in Theorem 4.1 with respect to depth. We construct four MLPs of depths: 5, 7, 9, and 11, trained on the MNIST dataset with architectures provided in the Appendix C. Independently of the number of bits, the ratio value is dependent on the depth, starting from $\approx 10^3$ for depth 5 to $\approx 10^8$ for depth 11. Additionally, it is observed that for extreme quantization (4 bits), there is a notable difference compared to other quantizations (8, 16, and 24 bits). Indeed, for 4 bits, the model accuracy is bad ($\leq 10\%$), impacting the norms of the weight matrices and making $r_{mean}$ closer to $r_{max}$. This results in a slightly lower ratio compared to other quantization levels. Finally, in Figure 5b, we compare the bounds with respect to the approximation error. First, the slope of the curves does not change. Moreover, the exponential dependence on depth is significantly reduced. Indeed, the slope given by $r$ for the previous bound is much steeper than the slope given by $r_{mean}$ in our bound. Thus, the new bound better adapts to the network's behavior, up to a constant. However, the ideal slope, which would perfectly match the network's behavior, would be a horizontal curve. We extended our experiments on MLPs by replacing the ReLU activation with a sigmoid function implemented with the `tanh`. This function satisfies our hypotheses and the experimental results show similar trends to those obtained with ReLU, showing that our approach is robust to different activation functions (see Figure 12 in Appendix I).

## 6 Conclusion and Perspectives

In this work, we introduced a novel theoretical approximation bound for deep (convolutional) neural networks. By generalizing the infinity-norm-based bound and introducing a more flexible approach to operator norm constraints, our bound significantly improves existing ones for a broad range of network architectures. This was

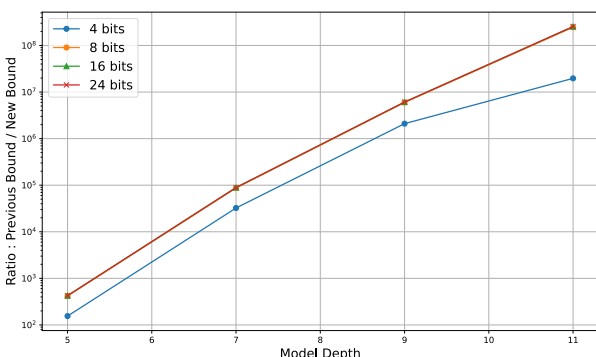

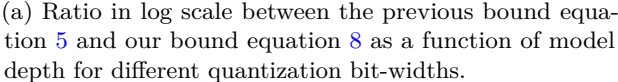

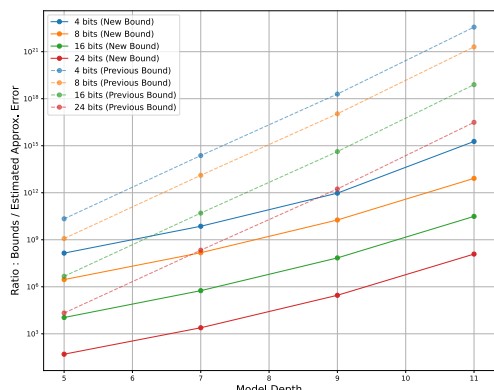

(a) Ratio in log scale between the previous bound equation 5 and our bound equation 8 as a function of model depth for different quantization bit-widths.

(b) Ratio in log scale between bounds equation 5, equation 8 and the estimated error approximation, as a function of model depth for different quantization bit-widths.

Figure 5: Comparison for MLPs of depths 5, 7, 9 and 11 on MNIST. (a) shows how the ratio of our bound over the previous bound grows exponentially with depth, and (b) demonstrates that our bound reduces that exponential dependence across bit-widths.

validated on popular architectures such as ResNet18, ResNet50, and MobileNetV2, as well as MLPs without bias quantization; showcasing improved accuracy in performance predictions for post-training quantized networks. The present study is focused on CNNs, and extensions to other architectures are beyond the scope of this paper. It is nevertheless clear that extending our theoretical framework to Vision Transformers (Dosovitskiy et al., 2020) would be highly interesting for future work. Such an extension would require a careful treatment of specific Transformers block, including layer normalization and multi-head attention.

As for layer normalization, it is not globally Lipschitz due to its input-dependent scaling. However, after training, one can empirically estimate a Lipschitz constant $L_{\mathrm{LN}}$ for each layer, which can then be integrated into our framework in the same way as batch normalization, as discussed in Section 5.

Regarding multi-head attention, the main difficulty lies in bounding the softmax operation used in scaled dot-product attention:

$$\mathrm{Att}(Q, K, V) = \mathrm{softmax}\left(\frac{QK^\top}{\sqrt{d}}\right) V.$$

While softmax is not globally 1-Lipschitz, it is locally Lipschitz over bounded domains. A local Lipschitz constant $L_{\mathrm{soft}}$ can be extracted depending on the norm of the inputs, and included in the bound. Recent work (Castin et al., 2023) provides precise estimates for this constant. The concatenation of multiple attention heads can be handled by bounding each head independently and summing their contributions, although this may result in pessimistic bounds. Skip connections can be addressed on a case-by-case basis depending on the model architecture, as we did for CNNs.

Another important and promising lead lies in developing a probabilistic approach that reflects the network's expected behavior under typical operating conditions, such as average or quantile-based performance, providing a complementary view to our deterministic bound. While a deterministic bound is crucial for critical applications, probabilistic insights can enhance our understanding of the network's behavior in practical, real-world scenarios.

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

# A  Proof of main results

For completeness, we begin by recalling the value of the infinity norm of a matrix.

**Lemma A.1.** *For all matrices $A \in \mathbb{R}^{m \times n}$ and all $x \in \mathbb{R}^n$ we have:*

$$\|A\|_{\mathrm{op},\infty} := \sup_{x \in \mathbb{R}^n, \, \|x\|_\infty = 1} \|Wx\|_\infty = \max_{1 \leq i \leq m} \sum_{j=1}^n |a_{i,j}| \tag{20}$$

*Proof.* Let $A \in \mathbb{R}^{m \times n}$ and $x \in \mathbb{R}^n$ with $\|x\|_\infty = 1$, then

$$\|Ax\|_\infty = \max_{1 \leq i \leq m} |(Ax)_i| = \max_{1 \leq i \leq m} \left| \sum_{j=1}^n a_{i,j} x_j \right| \leq \max_{1 \leq i \leq m} \sum_{j=1}^n |a_{i,j}| \, \|x\|_\infty = \max_{1 \leq i \leq m} \sum_{j=1}^n |a_{i,j}| \tag{21}$$

To show that equality holds, let $x$ such that $x_j = \mathrm{sign}(a_{i^\star,j})$ where $i^\star \in \arg\max_i \sum_{j=1}^n |a_{i,j}|$. Then $\|x\|_\infty = 1$ and:

$$\|Ax\|_\infty = \sum_{j=1}^n a_{i^\star j} \cdot \mathrm{sign}(a_{i^\star j}) = \sum_{j=1}^n |a_{i^\star j}| = \max_i \sum_{j=1}^n |a_{i,j}|. \tag{22}$$

This shows that:

$$\|A\|_{\mathrm{op},\infty} = \max_{1 \leq i \leq m} \sum_{j=1}^n |a_{ij}|. \tag{23}$$

$\square$

The following Lemma bounds the difference of outputs between a network and its quantization. It is adapted from [Lemma C.1]Gonon et al. (2023) to weight matrices with included identical biases.

**Lemma A.2.** *Let $(L, \mathbf{N})$ be an architecture with $L \geq 1$, denoting $\theta = (\tilde{W}_1, \ldots, \tilde{W}_L)$, $\theta' = (\tilde{W}'_1, \ldots, \tilde{W}'_L) \in \Theta_{L,\mathbf{N}}$ as two sets of parameters associated with this architecture, with each last column of the matrices composed by biases of the corresponding layer. We assume that the two networks have same biases (i.e. we do not quantize the bias) For every $\ell = 1, \ldots, L-1$, define $\theta'_\ell$ as the parameter deduced from $\theta'$, associated with the architecture $(\ell, (N_0, \ldots, N_\ell))$:*

$$\theta'_\ell = (\tilde{W}'_1, \ldots, \tilde{W}'_\ell).$$

*Then for every $\tilde{x} = \begin{pmatrix} x \\ 1 \end{pmatrix}$ with $x \in \mathbb{R}^{N_0}$, denoting by $W_k$ and $W'_k$ the weight matrices without biases (i.e. $\tilde{W}_k$ and $\tilde{W}'_k$ with last column removed), then, for any 1-Lipschitz activation function $\sigma$, we have:*

$$\|R_\theta(\tilde{x}) - R_{\theta'}(\tilde{x})\|_\infty \leq \sum_{\ell=1}^L \left( \prod_{k=\ell+1}^L \|W_k\|_{\mathrm{op},\infty} \right) \|W_\ell - W'_\ell\|_{\mathrm{op},\infty} \|R_{\theta'_{\ell-1}}(\tilde{x})\|_\infty, \tag{24}$$

*where we set by convention $R_{\theta'_{\ell-1}}(\tilde{x}) = x$ for $\ell = 1$, and $\prod_{k=\ell+1}^L \|W_k\|_{\mathrm{op},\infty} = 1$ for $\ell = L$.*

*Proof.* The proof of Inequality equation 24 follows by induction on $L \in \mathbb{N}$. For $L = 1$, with the Lipschitz condition and the definition of $\|\cdot\|_{\mathrm{op},\infty}$, using the fact that the last columns of $\tilde{W}_1$ and $\tilde{W}'_1$ are equal, we have $\|\tilde{W}_1 \tilde{x} - \tilde{W}'_1 \tilde{x}\|_\infty = \|W_1 x - W'_1 x\|_\infty$ and:

$$\begin{aligned} \left\| R_{\theta_L}(\tilde{x}) - R_{\theta'_L}(\tilde{x}) \right\|_\infty &= \left\| \sigma(\tilde{W}_1 \tilde{x}) - \sigma(\tilde{W}'_1 \tilde{x}) \right\|_\infty \leq \left\| \tilde{W}_1 \tilde{x} - \tilde{W}'_1 \tilde{x} \right\|_\infty \\ &= \|W_1 \tilde{x} - W'_1 x\|_\infty \\ &\leq \|W_1 - W'_1\|_{\mathrm{op},\infty} \|x\|_\infty. \end{aligned} \tag{25}$$

Now assume that property equation 24 holds for $L \geq 1$. At rank $L + 1$, using the fact that the activation function $\sigma$ is 1-Lipschitz, we have :

$$
\begin{aligned}
&\left\| R_{\theta_{L+1}}(\tilde{x}) - R_{\theta'_{L+1}}(\tilde{x}) \right\|_\infty \\
&= \left\| \sigma \left( \tilde{W}_{L+1} \begin{pmatrix} R_{\theta_L}(\tilde{x}) \\ 1 \end{pmatrix} \right) - \sigma \left( \tilde{W}'_{L+1} \begin{pmatrix} R_{\theta'_L}(\tilde{x}) \\ 1 \end{pmatrix} \right) \right\|_\infty & & \Big) \text{ } \sigma \text{ is 1-Lipschitz} \\
&\leq \left\| \tilde{W}_{L+1} \begin{pmatrix} R_{\theta_L}(\tilde{x}) \\ 1 \end{pmatrix} - \tilde{W}'_{L+1} \begin{pmatrix} R_{\theta'_L}(\tilde{x}) \\ 1 \end{pmatrix} \right\|_\infty \\
&= \left\| \tilde{W}_{L+1} \left( \begin{pmatrix} R_{\theta_L}(\tilde{x}) \\ 1 \end{pmatrix} - \begin{pmatrix} R_{\theta'_L}(\tilde{x}) \\ 1 \end{pmatrix} \right) + (\tilde{W}_{L+1} - \tilde{W}'_{L+1}) \begin{pmatrix} R_{\theta'_L}(\tilde{x}) \\ 1 \end{pmatrix} \right\|_\infty & & \Big) \text{ } \textit{triangle inequality} \\
&\leq \left\| \tilde{W}_{L+1} \begin{pmatrix} R_{\theta_L}(\tilde{x}) - R_{\theta'_L}(\tilde{x}) \\ 0 \end{pmatrix} \right\|_\infty + \left\| (\tilde{W}_{L+1} - \tilde{W}'_{L+1}) \begin{pmatrix} R_{\theta'_L}(\tilde{x}) \\ 1 \end{pmatrix} \right\|_\infty & & \Big) \text{ } \textit{same last column} \\
&= \left\| W_{L+1} \left( R_{\theta_L}(\tilde{x}) - R_{\theta'_L}(\tilde{x}) \right) \right\|_\infty + \left\| (W_{L+1} - W'_{L+1}) R_{\theta'_L}(\tilde{x}) \right\|_\infty & & \text{ } \textit{for } \tilde{W}_{L+1} \textit{ and } \tilde{W}'_{L+1} \\
&\leq \| W_{L+1} \|_{\mathrm{op},\infty} \left\| R_{\theta_L}(\tilde{x}) - R_{\theta'_L}(\tilde{x}) \right\|_\infty + \| W_{L+1} - W'_{L+1} \|_{\mathrm{op},\infty} \left\| R_{\theta'_L}(\tilde{x}) \right\|_\infty. & & \Big) \text{ } \textit{Sub-multiplicativity}
\end{aligned}
$$

Applying the induction hypothesis (Inequality equation 24) to the term $\left\| R_{\theta_L}(\tilde{x}) - R_{\theta'_L}(\tilde{x}) \right\|_\infty$, we have:

$$
\left\| R_{\theta_L}(\tilde{x}) - R_{\theta'_L}(\tilde{x}) \right\|_\infty \leq \sum_{\ell=1}^{L} \left( \prod_{k=\ell+1}^{L} \| W_k \|_{\mathrm{op},\infty} \right) \| W_\ell - W'_\ell \|_{\mathrm{op},\infty} \left\| R_{\theta'_{\ell-1}}(\tilde{x}) \right\|_\infty. \tag{26}
$$

Substituting this bound back into the previous inequality and using that we have $\prod_{k=L+2}^{L=1} \| W_k \|_{\mathrm{op},\infty} = 1$ by convention, we get:

$$
\begin{aligned}
\left\| R_{\theta_{L+1}}(\tilde{x}) - R_{\theta'_{L+1}}(\tilde{x}) \right\|_\infty &\leq \| W_{L+1} \|_{\mathrm{op},\infty} \sum_{\ell=1}^{L} \left( \prod_{k=\ell+1}^{L} \| W_k \|_{\mathrm{op},\infty} \right) \| W_\ell - W'_\ell \|_{\mathrm{op},\infty} \left\| R_{\theta'_{\ell-1}}(\tilde{x}) \right\|_\infty \\
&\quad + \| W_{L+1} - W'_{L+1} \|_{\mathrm{op},\infty} \left\| R_{\theta'_L}(\tilde{x}) \right\|_\infty \\
&= \sum_{\ell=1}^{L} \left( \prod_{k=\ell+1}^{L+1} \| W_k \|_{\mathrm{op},\infty} \right) \| W_\ell - W'_\ell \|_{\mathrm{op},\infty} \left\| R_{\theta'_{\ell-1}}(\tilde{x}) \right\|_\infty \\
&\quad + \left( \prod_{k=L+1+1}^{L+1} \| W_k \|_{\mathrm{op},\infty} \right) \| W_{L+1} - W'_{L+1} \|_{\mathrm{op},\infty} \left\| R_{\theta'_L}(\tilde{x}) \right\|_\infty.
\end{aligned} \tag{27}
$$

We deduce

$$
\left\| R_{\theta_{L+1}}(\tilde{x}) - R_{\theta'_{L+1}}(\tilde{x}) \right\|_\infty \leq \sum_{\ell=1}^{L+1} \left( \prod_{k=\ell+1}^{L+1} \| W_k \|_{\mathrm{op},\infty} \right) \| W_\ell - W'_\ell \|_{\mathrm{op},\infty} \left\| R_{\theta'_{\ell-1}}(\tilde{x}) \right\|_\infty. \tag{28}
$$

This concludes the induction and proves the lemma.

$\square$

Note that, if we suppose that our networks have no bias, then, for all layers, we do not have to take account of the last column of weight matrices with included bias. We can do the same proof replacing $\tilde{W}_l$ by $W_l$ and $\tilde{x}$ by $x$.

The result without bias also follows by the same induction, and it comes:

$$\|R_\theta(x) - R_{\theta'}(x)\|_\infty \leq \sum_{\ell=1}^{L} \left( \prod_{k=\ell+1}^{L} \|W_k\|_{\text{op},\infty} \right) \|W_\ell - W'_\ell\|_{\text{op},\infty} \left\| R_{\theta'_{\ell-1}}(x) \right\|_\infty. \tag{29}$$

The following lemma allows us to upper bound the output of a network based on the operator norms of each layer, without any specific conditions on their values.

**Lemma A.3.** *Let* $(L, \mathbf{N})$ *be an architecture with* $L \geq 1$, *denoting* $\theta = (\tilde{W}_1, \ldots, \tilde{W}_L) \in \Theta_{L,\mathbf{N}}$ *a set of parameters associated with this architecture, adding the assumption* $\sigma(0) = 0$. *Then for every* $\tilde{x} = \begin{pmatrix} x \\ 1 \end{pmatrix}$ *where* $x \in \mathbb{R}^{N_0}$, *we have:*

$$\|R_{\theta_L}(\tilde{x})\|_\infty \leq \max \left( \max_{l=2,\ldots,L} \prod_{s=\ell}^{L} \|\tilde{W}_s\|_{\text{op},\infty}; \prod_{s=1}^{L} \|\tilde{W}_s\|_{\text{op},\infty} \|\tilde{x}\|_\infty \right). \tag{30}$$

*Proof.* We prove equation 30 by induction on $L \in \mathbb{N}$. For $L = 1$, since $\sigma$ is 1-Lipschitz and satisfies $\sigma(0) = 0$, we have:

$$\|R_{\theta_1}(\tilde{x})\|_\infty = \|\sigma(\tilde{W}_1 \tilde{x}) - \sigma(0)\|_\infty \leq \|\tilde{W}_1 \tilde{x} - 0\|_\infty = \|\tilde{W}_1 \tilde{x}\|_\infty \leq \|\tilde{W}_1\|_{\text{op},\infty} \|\tilde{x}\|_\infty. \tag{31}$$

Then by convention for $L = 1$ (as in Lemma A.2),

$$\max_{l=2,\ldots,L} \prod_{s=\ell}^{L} \|\tilde{W}_s\|_{\text{op},\infty} = 1. \tag{32}$$

We deduce:

$$\|R_{\theta_L}(\tilde{x})\|_\infty \leq \|\tilde{W}_1\|_{\text{op},\infty} \|\tilde{x}\|_\infty \leq \max(1; \|\tilde{W}_1\|_{\text{op},\infty} \|\tilde{x}\|_\infty). \tag{33}$$

Now assume that the property holds for $L \geq 1$. At rank $L + 1$, using that operator norm is sub-multiplicative and the fact that $\sigma$ is 1-Lipschitz and satisfies $\sigma(0) = 0$, we have:

$$\begin{aligned}
\|R_{\theta_{L+1}}(\tilde{x})\|_\infty &= \left\| \sigma \left( \tilde{W}_{L+1} \begin{pmatrix} R_{\theta_L}(\tilde{x}) \\ 1 \end{pmatrix} \right) \right\|_\infty \\
&\leq \|\tilde{W}_{L+1}\|_{\text{op},\infty} \left\| \begin{pmatrix} R_{\theta_L}(\tilde{x}) \\ 1 \end{pmatrix} \right\|_\infty \\
&= \left\| \begin{pmatrix} \|\tilde{W}_{L+1}\|_{\text{op},\infty} \|R_{\theta_L}(\tilde{x})\|_\infty \\ \|\tilde{W}_{L+1}\|_{\text{op},\infty} \end{pmatrix} \right\|_\infty
\end{aligned} \tag{34}$$

Then, applying the induction hypothesis to the term $\|R_{\theta_L}(\tilde{x})\|_\infty$, it comes:

$$\left\| \begin{pmatrix} \|\tilde{W}_{L+1}\|_{\mathrm{op},\infty} \|R_{\theta_L}(\tilde{x})\|_\infty \\ \|\tilde{W}_{L+1}\|_{\mathrm{op},\infty} \end{pmatrix} \right\|_\infty = \max\left( \|\tilde{W}_{L+1}\|_{\mathrm{op},\infty} \|R_{\theta_L}(\tilde{x})\|_\infty; \|\tilde{W}_{L+1}\|_{\mathrm{op},\infty} \right)$$

$$\leq \max\left( \|\tilde{W}_{L+1}\|_{\mathrm{op},\infty} \cdot \max\left[ \max_{l=2,\dots,L} \prod_{s=l}^{L} \|\tilde{W}_s\|_{\mathrm{op},\infty}; \right.\right.$$

$$\left.\left. \prod_{s=1}^{L} \|\tilde{W}_s\|_{\mathrm{op},\infty} \|\tilde{x}\|_\infty \right]; \|\tilde{W}_{L+1}\|_{\mathrm{op},\infty} \right)$$

$$= \max\left( \max\left[ \max_{l=2,\dots,L} \prod_{s=l}^{L+1} \|\tilde{W}_s\|_{\mathrm{op},\infty}; \right.\right. \tag{35}$$

$$\left.\left. \prod_{s=1}^{L+1} \|\tilde{W}_s\|_{\mathrm{op},\infty} \|\tilde{x}\|_\infty \right]; \|\tilde{W}_{L+1}\|_{\mathrm{op},\infty} \right)$$

$$= \max\left[ \max_{l=2,\dots,L} \prod_{s=l}^{L+1} \|\tilde{W}_s\|_{\mathrm{op},\infty}; \right.$$

$$\left. \prod_{s=1}^{L+1} \|\tilde{W}_s\|_{\mathrm{op},\infty} \|\tilde{x}\|_\infty; \|\tilde{W}_{L+1}\|_{\mathrm{op},\infty} \right]$$

Then, noticing that the term $\|\tilde{W}_{L+1}\|_{\mathrm{op},\infty}$ can be included in $\max_{l=2,\dots,L}\left(\prod_{s=l}^{L+1}\|\tilde{W}_s\|_{\mathrm{op},\infty}\right)$ by adding index $\ell = L+1$, we have

$$\|R_{\theta_{L+1}}(\tilde{x})\|_\infty \leq \max\left[ \max_{l=2,\dots,L+1} \left( \prod_{s=l}^{L+1} \|\tilde{W}_s\|_{\mathrm{op},\infty} \right); \prod_{s=1}^{L+1} \|\tilde{W}_s\|_{\mathrm{op},\infty} \|\tilde{x}\|_\infty \right]. \tag{36}$$

This concludes the induction and proves the lemma.

$\square$

We can now prove our main theorem.

*Proof of Theorem 4.1.* We recall that $\theta'_\ell$ is defined as the parameter deduced from $\theta'$, associated with the architecture $(\ell, (N_0, \dots, N_\ell))$.

With the convention that

$$\begin{cases} R_{\theta'_{l-1}}(\tilde{x}) = \tilde{x} & \text{if } l = 1, \\ \prod_{k=\ell+1}^{L} \|W_k\|_{\mathrm{op},q} = 1 & \text{if } l = L, \end{cases} \tag{37}$$

we have, with Lemma A.2:

$$\|R_\theta(\tilde{x}) - R_{\theta'}(\tilde{x})\|_\infty \leq \sum_{\ell=1}^{L} \left( \prod_{k=\ell+1}^{L} \|W_k\|_{\mathrm{op},\infty} \right) \|W_\ell - W'_\ell\|_{\mathrm{op},\infty} \|R_{\theta'_{\ell-1}}(\tilde{x})\|_\infty. \tag{38}$$

Thus, using Lemma A.3 we can bound $\|R_{\theta'_{\ell-1}}(\tilde{x})\|_\infty$, with the convention that an empty product is equal to 1, it follows:

$$\|R_{\theta'_{\ell-1}}(\tilde{x})\|_\infty \le \max\left[ \max_{i=2,\dots,l-1}\left(\prod_{s=i}^{l-1}\|\tilde{W}'_s\|_{\mathrm{op},\infty}\right); \prod_{s=1}^{l-1}\|\tilde{W}'_s\|_{\mathrm{op},\infty}\|\tilde{x}\|_\infty \right] \tag{39}$$

Then, noticing that $\|\tilde{x}\|_\infty \ge 1$, and re-indexing the second max to include $\prod_{s=1}^{l-1}\|\tilde{W}'_s\|_{\mathrm{op},\infty}$, we have:

$$
\begin{aligned}
\|R_{\theta'_{\ell-1}}(\tilde{x})\|_\infty &\le \max\left[ \max_{i=2,\dots,l-1}\left(\prod_{s=i}^{l-1}\|\tilde{W}'_s\|_{\mathrm{op},\infty}\right)\|\tilde{x}\|_\infty; \prod_{s=1}^{l-1}\|\tilde{W}'_s\|_{\mathrm{op},\infty}\|\tilde{x}\|_\infty \right] \\
&= \|\tilde{x}\|_\infty \max\left[ \max_{i=2,\dots,l-1}\left(\prod_{s=i}^{l-1}\|\tilde{W}'_s\|_{\mathrm{op},\infty}\right); \prod_{s=1}^{l-1}\|\tilde{W}'_s\|_{\mathrm{op},\infty} \right] \\
&= \|\tilde{x}\|_\infty \max_{i=1,\dots,l-1}\left(\prod_{s=i}^{l-1}\|\tilde{W}'_s\|_{\mathrm{op},\infty}\right).
\end{aligned}
\tag{40}
$$

Using this bound in Equation equation 38, we get:

$$\|R_\theta(\tilde{x}) - R_{\theta'}(\tilde{x})\|_\infty \le \|\tilde{x}\|_\infty \sum_{\ell=1}^{L}\left(\prod_{k=\ell+1}^{L}\|W_k\|_{\mathrm{op},\infty}\right)\|W_\ell - W'_\ell\|_{\mathrm{op},\infty} \max_{i=1,\dots,l-1}\left(\prod_{s=i}^{l-1}\|\tilde{W}'_s\|_{\mathrm{op},\infty}\right). \tag{41}$$

Then, recalling that $\tilde{x} = \begin{pmatrix} x \\ 1 \end{pmatrix}$ with $x \in [-D, D]^d$, we have

$$\|\tilde{x}\|_\infty \le \max(D; 1) \tag{42}$$

Then, we know by Lemma A.1 that for every matrix in $\mathbb{R}^{m \times n}$:

$$\|W\|_{\mathrm{op},\infty} = \max_{1 \le i \le m} \sum_{j=1}^{n} |w_{ij}| \tag{43}$$

Thus, recalling that $\theta = (\tilde{W}_1, \dots, \tilde{W}_L)$ and because for all $l$, $dim(W_l) = N_l \times N_{l-1}$, we can write:

$$\|W_l\|_{\mathrm{op},\infty} \le N_{l-1} \max_{i,j}|(W_l)_{ij}| \le N_{l-1}\|\theta\|_\infty \tag{44}$$

Using the previous inequality on $\|W_l - W'_l\|_{\mathrm{op},\infty}$ and replacing it in Inequality equation 41, we deduce that:

$$\|R_\theta(\tilde{x}) - R_{\theta'}(\tilde{x})\|_\infty \le \max(D; 1) \sum_{\ell=1}^{L} N_{l-1}\left(\prod_{k=\ell+1}^{L}\|W_k\|_{\mathrm{op},\infty}\right)\max_{i=1,\dots,l-1}\left(\prod_{s=i}^{l-1}\|\tilde{W}'_s\|_{\mathrm{op},\infty}\right)\|\theta - \theta'\|_\infty \tag{45}$$

Thus, recalling that for all $l$,

$$\|\tilde{W}_l\|_{\mathrm{op},\infty} \le r_l \quad \text{and} \quad \|\tilde{W}'_l\|_{\mathrm{op},\infty} \le r_l \tag{46}$$

Equation equation 45 becomes:

$$\|R_\theta(\tilde{x}) - R_{\theta'}(\tilde{x})\|_\infty \leq \max(D;1) \sum_{\ell=1}^{L} N_{l-1} \left( \prod_{k=\ell+1}^{L} r_k \right) \max_{i=1,\ldots,l-1} \left( \prod_{s=i}^{l-1} r_s \right) \|\theta - \theta'\|_\infty$$

$$\leq \max(D;1) \sum_{\ell=1}^{L} \left( N_{l-1} \max_{i=1,\ldots,l-1} \prod_{\substack{j=i \\ j\neq l}}^{L} r_j \right) \|\theta - \theta'\|_\infty \tag{47}$$

Then taking the maximum over all layers, we finally have:

$$\|R_\theta(\tilde{x}) - R_{\theta'}(\tilde{x})\|_\infty \leq \max(D;1) \left( \max_{l=1,\ldots,L} \max_{i=1,\ldots,l-1} \prod_{\substack{j=i \\ j\neq l}}^{L} r_j \right) \sum_{\ell=1}^{L} N_{l-1} \|\theta - \theta'\|_\infty \tag{48}$$

Then, we can rewrite this to show the geometric mean for partial products:

$$\|R_\theta(\tilde{x}) - R_{\theta'}(\tilde{x})\|_\infty \leq \max(D;1) \left( \sqrt[L-1]{\max_{l=1,\ldots,L} \left( \max_{i=1,\ldots,l-1} \prod_{\substack{j=i \\ j\neq l}}^{L} r_j \right)} \right)^{L-1} \sum_{\ell=1}^{L} N_{\ell-1} \|\theta - \theta'\|_\infty \tag{49}$$

Finally, taking the supremum of both sides we obtain:

$$\sup_{x\in\Omega} \|R_\theta(\tilde{x}) - R_{\theta'}(\tilde{x})\|_\infty \leq \max(D;1) \left( \sqrt[L-1]{\max_{l=1,\ldots,L} \left( \max_{i=1,\ldots,l-1} \prod_{\substack{j=i \\ j\neq l}}^{L} r_j \right)} \right)^{L-1} \sum_{\ell=1}^{L} N_{\ell-1} \|\theta - \theta'\|_\infty \tag{50}$$

$\square$

We now provide an intermediate Theorem (dealing with MLP without biases) to complete Theorem 4.1 (MLP with biases not quantified) and Theorem 4.4 (CNN with no biases).

**Theorem A.4** (Bound for neural networks without bias). *With the same settings as in Theorem 4.1, with* $(b_1,\ldots,b_L) = (b'_1,\ldots,b'_L) = (0,\ldots,0)$ *(i.e $\forall l$, we can take $\tilde{W}_l = W_l$ and $\tilde{W}'_l = W'_l$, the standard weight matrices without included bias). For all $x \in \Omega$ the following bound holds.*

$$\sup_{x\in\Omega} \|R_\theta(x) - R_{\theta'}(x)\|_\infty \leq D \left( \sqrt[L-1]{\max_{l=1,\ldots,L} \prod_{\substack{k=1 \\ k\neq l}}^{L} r_k} \right)^{L-1} \sum_{\ell=1}^{L} N_{l-1} \|\theta - \theta'\|_\infty. \tag{51}$$

*Proof.* By analogy of the previous proof, we use Equation equation 29 (which corresponds to Lemma A.2 but without bias), and it comes:

$$\|R_\theta(x) - R_{\theta'}(x)\|_\infty \leq \sum_{\ell=1}^{L} \left( \prod_{k=\ell+1}^{L} \|W_k\|_{\mathrm{op},\infty} \right) \|W_\ell - W'_\ell\|_{\mathrm{op},\infty} \left\| R_{\theta'_{\ell-1}}(x) \right\|_\infty \tag{52}$$

Now we want to bound the term $\left\|R_{\theta'_{\ell-1}}(x)\right\|_\infty$ by using equation 29. Thus, for any $\theta$ and with $\theta' = (0, \ldots, 0)$, noticing that:

$$\forall l \geq 2, \ \|R_{\theta'_{\ell-1}}(x)\|_\infty = 0,$$
$$\text{if } l = 1, \ \|R_{\theta'_{\ell-1}}(x)\|_\infty = \|x\|_\infty, \text{by convention} \tag{53}$$

We have:

$$\|R_{\theta_{l-1}}(x)\|_\infty \leq \prod_{k=2}^{l-1} \|W_k\|_{\text{op},\infty} \|W_1 - 0\|_{\text{op},\infty} \|x\|_\infty = \prod_{k=1}^{l-1} \|W_k\|_{\text{op},\infty} \|x\|_\infty \tag{54}$$

Now for any $\theta, \theta'$ without bias, we can bound $\left\|R_{\theta'_{\ell-1}}(x)\right\|_\infty$ in equation 29 , and it comes:

$$\|R_\theta(x) - R_{\theta'}(x)\|_\infty \leq \|x\|_\infty \sum_{\ell=1}^{L} \left(\prod_{k=\ell+1}^{L} \|W_k\|_{\text{op},\infty}\right) \prod_{k=1}^{l-1} \|W'_k\|_{\text{op},\infty} \|W_\ell - W'_\ell\|_{\text{op},\infty} \tag{55}$$

Then, we use Lemma A.1 to get:

$$\|W_l\|_{\text{op},\infty} \leq N_{l-1} \max_{i,j} |(W_l)_{ij}| \leq N_{l-1} \|\theta\|_\infty \tag{56}$$

Hence recalling that for all $k$ we have $\|W_k\|_{\text{op},\infty} \leq r_k$ and $\|W'_k\|_{\text{op},\infty} \leq r_k$, it comes:

$$\|R_\theta(x) - R_{\theta'}(x)\|_\infty \leq \|x\|_\infty \sum_{\ell=1}^{L} N_{l-1} \left(\prod_{k=\ell+1}^{L} \|W_k\|_{\text{op},\infty}\right) \prod_{k=1}^{l-1} \|W'_k\|_{\text{op},\infty} \|\theta - \theta'\|_\infty$$
$$\leq D \sum_{\ell=1}^{L} N_{l-1} \prod_{\substack{k=1 \\ k \neq l}}^{L} r_k \|\theta - \theta'\|_\infty \tag{57}$$

Thus taking the maximum over all layers, we can rewrite the bound in terms of the geometric mean:

$$\|R_\theta(x) - R_{\theta'}(x)\|_\infty \leq D \left(\sqrt[L-1]{\max_{l=1,\ldots,L} \prod_{\substack{k=1 \\ k \neq l}}^{L} r_k}\right)^{L-1} \sum_{\ell=1}^{L} N_{l-1} \|\theta - \theta'\|_\infty \tag{58}$$

Finally, taking the supremum gives the desired result:

$$\sup_{x \in \Omega} \|R_\theta(x) - R_{\theta'}(x)\|_\infty \leq D \left(\sqrt[L-1]{\max_{l=1,\ldots,L} \prod_{\substack{k=1 \\ k \neq l}}^{L} r_k}\right)^{L-1} \sum_{\ell=1}^{L} N_{l-1} \|\theta - \theta'\|_\infty. \tag{59}$$

$\square$

We now give the proof of our last theorem.

*Proof of Theorem 4.4.* Let $\theta = (\mathcal{H}_1, \ldots, \mathcal{H}_L)$ the vector of parameters where each $\mathcal{H}_\ell$ represent the convolution matrix of layer $l$.

With the convention that

$$\begin{cases} R_{\theta'_{l-1}}(x) = x & \text{if } l = 1, \\ \prod_{k=\ell+1}^{L} \|\mathcal{H}_k\|_{\text{op},\infty} = 1 & \text{if } l = L \end{cases} \tag{60}$$

We can start the proof using the same line of reasoning. Thus we can use directly Equation equation 55 that becomes for the convolutional case:

$$\|R_\theta(x) - R_{\theta'}(x)\|_\infty \leq \|x\|_\infty \sum_{l=1}^{L} \prod_{k=\ell+1}^{L} \|\mathcal{H}_k\|_{\text{op},\infty} \prod_{k=1}^{l-1} \|\mathcal{H}'_k\|_{\text{op},\infty} \times \|\mathcal{H}_l - \mathcal{H}'_l\|_{\text{op},\infty} \tag{61}$$

Then by analogy we can write:

$$\begin{aligned}
\|R_\theta(x) - R_{\theta'}(x)\|_\infty &\leq \|x\|_\infty \sum_{l=1}^{L} \prod_{k=\ell+1}^{L} \|\mathcal{H}_k\|_{\text{op},\infty} \prod_{k=1}^{l-1} \|\mathcal{H}'_k\|_{\text{op},\infty} \times \|\mathcal{H}_l - \mathcal{H}'_l\|_{\text{op},\infty} \\
&\leq D \sum_{l=1}^{L} \prod_{k=\ell+1}^{L} r_k \prod_{k=1}^{l-1} r_k \times \|\mathcal{H}_l - \mathcal{H}'_l\|_{\text{op},\infty} \\
&= D \sum_{l=1}^{L} \prod_{\substack{k=1 \\ k \neq l}}^{L} r_k \times \|\mathcal{H}_l - \mathcal{H}'_l\|_{\text{op},\infty}
\end{aligned} \tag{62}$$

Then, we know by lemma A.1 that for every matrix in $\mathbb{R}^{m \times n}$ it holds :

$$\|W\|_{\text{op},\infty} = \max_{1 \leq i \leq m} \sum_{j=1}^{n} |w_{ij}| \tag{63}$$

Thanks to the convolutional structure, we improve the bound on $\|\mathcal{H}_l\|_{\text{op},\infty}$ given by Lemma A.1. Let us note the output of the previous layer as $y_{l-1}$. This output is a set of feature maps with dimensions $(n_{l-1} \times m_{l-1}) \times c_{l-1}$, where $c_{l-1}$ is the number of feature maps (i.e., the number of filters) in the previous layer.

Next, recalling that, we want to express the convolution at layer $l$ as a matrix multiplication: $\mathcal{H}_l \text{vec}(y_{l-1})$.

Then we write $\mathcal{H}_l$ as a block-Toeplitz matrix:

$$\mathcal{H}_l = \begin{pmatrix} H_{l,1} \\ \vdots \\ H_{l,c_l} \end{pmatrix} \tag{64}$$

where each block $H_{l,i}$ is a Toeplitz matrix of size $(n_l m_l) \times (n_{l-1} m_{l-1} c_{l-1})$. These matrices are highly sparse, with each row composed of coefficients of the filters arranged in a specific pattern, while the remaining entries are zeros.

Thus, each block $H_{l,i}$ in $\mathcal{H}_l$ represents the convolution operation between the $i$-th filter and the feature maps of the previous layer.

Hence, the overall dimensions of $\mathcal{H}_l$ are:

$$\dim(\mathcal{H}_l) = (n_l m_l c_l) \times (n_{l-1} m_{l-1} c_{l-1}). \tag{65}$$

Then to bound the norm of $\mathcal{H}_l$, we consider only the non-zero coefficients in its rows, which are $p_l^2 \times c_{l-1}$, where $p_l^2$ denotes the size of the filters at layer $l$.

Thus, recalling that $\theta = (\mathcal{H}_1, \ldots, \mathcal{H}_L)$ we have:

$$\|\mathcal{H}_l\|_{\mathrm{op},\infty} \leq c_{l-1} \times p_l^2 \max_{i,j} |(\mathcal{H}_l)_{ij}| \leq c_{l-1} \times p_l^2 \|\theta\|_\infty \tag{66}$$

Using the previous inequality on $\|\mathcal{H}_l - \mathcal{H}_l'\|_{\mathrm{op},\infty}$ we can bound the quantity $\|R_\theta(x) - R_{\theta'}(x)\|_\infty$, by:

$$
\begin{aligned}
\|R_\theta(x) - R_{\theta'}(x)\|_\infty &\leq D \sum_{l=1}^{L} \prod_{\substack{k=1 \\ k \neq l}}^{L} r_k \times \|\mathcal{H}_l - \mathcal{H}_l'\|_{\mathrm{op},\infty} \\
&\leq D \sum_{l=1}^{L} c_{l-1} \times p_l^2 \prod_{\substack{k=1 \\ k \neq l}}^{L} r_k \|\theta - \theta'\|_\infty
\end{aligned} \tag{67}
$$

Thus taking the maximum over all layers, we can rewrite the bound in terms of the geometric mean:

$$\|R_\theta(x) - R_{\theta'}(x)\|_\infty \leq D \left( \sqrt[L-1]{\max_{l=1,\ldots,L} \prod_{\substack{k=1 \\ k \neq l}}^{L} r_k} \right)^{L-1} \sum_{\ell=1}^{L} c_{l-1} \times p_l^2 \|\theta - \theta'\|_\infty \tag{68}$$

Hence, we can conclude that:

$$\sup_{x \in \Omega} \|R_\theta(x) - R_{\theta'}(x)\|_\infty \leq D \left( \sqrt[L-1]{\max_{l=1,\ldots,L} \prod_{\substack{k=1 \\ k \neq l}}^{L} r_k} \right)^{L-1} \sum_{\ell=1}^{L} c_{l-1} \times p_l^2 \|\theta - \theta'\|_\infty. \tag{69}$$

$\square$

Then, we present the 3 specific architectures of residual and bottleneck block, in Resnet18, Resnet50 He et al. (2016) and MobilnetV2 Sandler et al. (2018).

# B   Calculations for MobileNetV2 and Resnets

## B.1   Specific structure of Resnet18

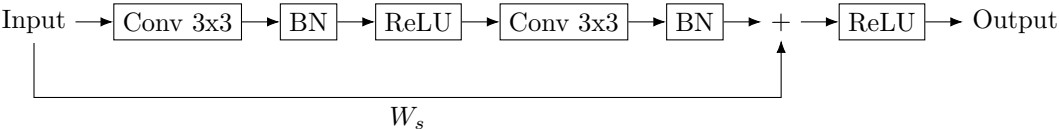

Figure 6: Structure of a Residual Block of ResNet18

**Lemma B.1** (Matrix Representation of a Residual Block in ResNet-18). *The output $y \in \mathbb{R}^n$ of a residual block in ResNet-18, as illustrated in Figure 6, can be expressed as:*

$$y = \sigma \left( V_2 \cdot \tilde{\sigma}_1 \left( V_1 \cdot f \right) \right), \tag{70}$$

*where $f \in \mathbb{R}^n$ is the input, and $V_1$ and $V_2$ are defined as:*

$$V_1 = \begin{pmatrix} W_1 \\ I \end{pmatrix} \in \mathbb{R}^{(d+n) \times n}, \quad V_2 = \begin{pmatrix} W_2 & W_s \end{pmatrix} \in \mathbb{R}^{m \times (d+n)}, \tag{71}$$

*with $W_1 \in \mathbb{R}^{d \times n}$ and $W_2 \in \mathbb{R}^{m \times d}$ as the convolutional weight matrices, $I \in \mathbb{R}^{n \times n}$ as the identity matrix and $W_s \in \mathbb{R}^{m \times n}$ represents the shortcut weight matrix. Note that, if the input and output of the block have the same dimension $W_s = I$. The function $\tilde{\sigma}_1$ applies the non-linearity $\sigma$ only to the term $W_1 \cdot f$, leaving the shortcut component $I \cdot f$ unchanged.*

*Proof.* The residual block computes the output $y$ as:

$$y = \sigma \left( \mathrm{BN}_2 \left( \mathrm{Conv}_2 \left( \mathrm{ReLU} \left( \mathrm{BN}_1 \left( \mathrm{Conv}_1(f) \right) \right) \right) \right) + W_s f \right), \tag{72}$$

where: $\mathrm{Conv}_1$ and $\mathrm{Conv}_2$ represent the convolutional layers with weight matrices $W_1$ and $W_2$, respectively, $\mathrm{BN}_1$ and $\mathrm{BN}_2$ are batch normalization layers (omitted in the matrix formulation because we removed them in our experiments).

The first convolutional layer computes:

$$x_1 = \mathrm{Conv}_1(f) = W_1 \cdot f, \tag{73}$$

where $W_1 \in \mathbb{R}^{d \times n}$. This result is passed through $\mathrm{BN}_1$ and $\sigma$, yielding:

$$\tilde{x}_1 = \sigma \left( \mathrm{BN}_1(x_1) \right). \tag{74}$$

In our matrix representation, we express this as:

$$\tilde{x}_1 = \tilde{\sigma}_1 \left( V_1 \cdot f \right), \tag{75}$$

where $V_1 = \begin{pmatrix} W_1 \\ I \end{pmatrix} \in \mathbb{R}^{(d+n) \times n}$. The expanded computation is:

$$V_1 \cdot f = \begin{pmatrix} W_1 \cdot f \\ I \cdot f \end{pmatrix} = \begin{pmatrix} W_1 \cdot f \\ f_1 \\ \vdots \\ f_n \end{pmatrix}. \tag{76}$$

Thus:

$$\tilde{\sigma}_1 \left( V_1 \cdot f \right) = \begin{pmatrix} \sigma(W_1 \cdot f) \\ f_1 \\ \vdots \\ f_n \end{pmatrix}. \tag{77}$$

Then the second convolutional layer computes:

$$x_2 = \mathrm{Conv}_2 \left( \sigma \left( \mathrm{BN}_1(x_1) \right) \right) = W_2 \cdot \tilde{x}_1, \tag{78}$$

where $W_2 \in \mathbb{R}^{m \times d}$. Combining the result with the shortcut connection $W_s f$, we obtain:

$$y = \sigma \left( x_2 + W_s f \right). \tag{79}$$

Using our matrix representation for $V_2$, where $V_2 = \begin{pmatrix} W_2 & W_s \end{pmatrix} \in \mathbb{R}^{m \times (d+n)}$, the computation expands as:

$$V_2 \cdot \begin{pmatrix} \sigma(W_1 \cdot f) \\ f_1 \\ \vdots \\ f_n \end{pmatrix} = \begin{pmatrix} W_2 & W_s \end{pmatrix} \cdot \begin{pmatrix} \sigma(W_1 \cdot f) \\ f_1 \\ \vdots \\ f_n \end{pmatrix} = W_2 \cdot \sigma(W_1 \cdot f) + W_s \cdot f = x_2 + W_s f. \tag{80}$$

So the final output of the block is:

$$y = \sigma\left(V_2 \cdot \tilde{\sigma}_1\left(V_1 \cdot f\right)\right), \tag{81}$$

Thus, the residual block's output is equivalent to our matrix representation. □

An immediate consequence of this Lemma is:

$$\begin{cases} \|V_1\|_{\mathrm{op},\infty} = \max(1, \|W_1\|_{\mathrm{op},\infty}), & \\ \|V_2\|_{\mathrm{op},\infty} = \|W_2\|_{\mathrm{op},\infty} + 1, & \text{if } W_s = I, \\ \|V_2\|_{\mathrm{op},\infty} \leq \|W_2\|_{\mathrm{op},\infty} + \|W_s\|_{\mathrm{op},\infty}, & \text{otherwise.} \end{cases} \tag{82}$$

## B.2 Specific structure of Resnet50

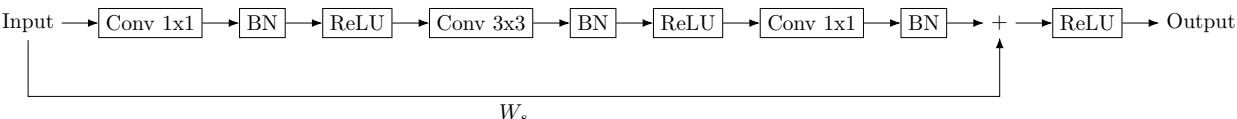

Figure 7: Structure of a Residual Block of ResNet50

**Lemma B.2** (Matrix Representation of a Bottleneck Block in ResNet-50). *The output $y \in \mathbb{R}^n$ of a bottleneck block in ResNet-50, as illustrated in Figure 7, can be expressed as:*

$$y = \sigma\left(V_3 \cdot \tilde{\sigma}_2\left(V_2 \cdot \tilde{\sigma}_1\left(V_1 \cdot f\right)\right)\right), \tag{83}$$

*where $f \in \mathbb{R}^n$ is the input, and the matrices $V_1$, $V_2$, and $V_3$ are defined as follows:*

$$V_1 = \begin{pmatrix} W_1 \\ I \end{pmatrix} \in \mathbb{R}^{(d_1+n) \times n}, \quad V_2 = \begin{pmatrix} W_2 & 0 \\ 0 & I \end{pmatrix} \in \mathbb{R}^{(d_2+n) \times (d_1+n)}, \quad V_3 = \begin{pmatrix} W_3 & W_s \end{pmatrix} \in \mathbb{R}^{m \times (d_2+n)}. \tag{84}$$

*Here: $W_1 \in \mathbb{R}^{d_1 \times n}$, $W_2 \in \mathbb{R}^{d_2 \times d_1}$, and $W_3 \in \mathbb{R}^{m \times d_2}$ are the weight matrices of the three convolutional layers in the bottleneck block, $W_s$ is the weight matrix associated with the shortcut, $I \in \mathbb{R}^{n \times n}$ is the identity, and $0$ denotes zero matrices of appropriate dimensions.*

*The functions $\tilde{\sigma}_1$ and $\tilde{\sigma}_2$ apply the non-linearity $\sigma$ only to specific components, leaving the shortcut components unchanged.*

*Proof.* The bottleneck block computes the output $y$ as:

$$y = \sigma\left(\mathrm{BN}_3\left(\mathrm{Conv}_3\left(\mathrm{ReLU}\left(\mathrm{BN}_2\left(\mathrm{Conv}_2\left(\mathrm{ReLU}\left(\mathrm{BN}_1\left(\mathrm{Conv}_1(f)\right)\right)\right)\right)\right)\right)\right) + W_s f\right), \tag{85}$$

where $\mathrm{Conv}_1$, $\mathrm{Conv}_2$, and $\mathrm{Conv}_3$ represent the three convolutional layers with weight matrices $W_1$, $W_2$, and $W_3$, respectively, $\mathrm{BN}_1$, $\mathrm{BN}_2$, and $\mathrm{BN}_3$ are batch normalization layers (omitted in the matrix formulation because we removed them in our experiments).

The first convolutional layer computes:

$$x_1 = \text{Conv}_1(f) = W_1 \cdot f, \tag{86}$$

where $W_1 \in \mathbb{R}^{d_1 \times n}$. This result is passed through $\text{BN}_1$ and $\sigma$, yielding:

$$\tilde{x}_1 = \sigma\left(\text{BN}_1(x_1)\right). \tag{87}$$

In our matrix representation, we express this as:

$$\tilde{x}_1 = \tilde{\sigma}_1\left(V_1 \cdot f\right), \tag{88}$$

where $V_1 = \begin{pmatrix} W_1 \\ I \end{pmatrix} \in \mathbb{R}^{(d_1+n) \times n}$. The expanded computation is:

$$V_1 \cdot f = \begin{pmatrix} W_1 \cdot f \\ I \cdot f \end{pmatrix} = \begin{pmatrix} W_1 \cdot f \\ f_1 \\ \vdots \\ f_n \end{pmatrix}. \tag{89}$$

Thus:

$$\tilde{\sigma}_1\left(V_1 \cdot f\right) = \begin{pmatrix} \sigma(W_1 \cdot f) \\ f_1 \\ \vdots \\ f_n \end{pmatrix}. \tag{90}$$

The second convolutional layer computes:

$$\tilde{x}_2 = \sigma(\text{Conv}_2(\tilde{x}_1)) = \sigma(W_2 \cdot \tilde{x}_1), \tag{91}$$

where $W_2 \in \mathbb{R}^{d_2 \times d_1}$. Using $V_2$, the expanded computation is:

$$V_2 \cdot \begin{pmatrix} \sigma(W_1 \cdot f) \\ f_1 \\ \vdots \\ f_n \end{pmatrix} = \begin{pmatrix} W_2 & 0 \\ 0 & I \end{pmatrix} \cdot \begin{pmatrix} \sigma(W_1 \cdot f) \\ f_1 \\ \vdots \\ f_n \end{pmatrix} = \begin{pmatrix} W_2 \cdot \sigma(W_1 \cdot f) \\ f_1 \\ \vdots \\ f_n \end{pmatrix}. \tag{92}$$

So the output of this layer with our matrix representation is: $\begin{pmatrix} \sigma(W_2 \cdot \sigma(W_1 \cdot f)) \\ f_1 \\ \vdots \\ f_n \end{pmatrix}.$

The third convolutional layer combines the outputs of the second layer and the shortcut connection:

$$x_3 = \text{Conv}_3(\tilde{x}_2) = W_3 \cdot \tilde{x}_2 + W_s f, \tag{93}$$

and the final original output of the block is $y = \sigma(x_3)$

Using $V_3$, this becomes:

$$V_3 \cdot \begin{pmatrix} \sigma(W_2 \cdot \sigma(W_1 \cdot f)) \\ f_1 \\ \vdots \\ f_n \end{pmatrix} = \begin{pmatrix} W_3 & W_s \end{pmatrix} \cdot \begin{pmatrix} \sigma(W_2 \cdot \sigma(W_1 \cdot f)) \\ f_1 \\ \vdots \\ f_n \end{pmatrix} = W_3 \cdot \sigma(W_2 \cdot \sigma(W_1 \cdot f)) + W_s f. \tag{94}$$

Then the final output is:

$$y = \sigma\left(V_3 \cdot \tilde{\sigma}_2\left(V_2 \cdot \tilde{\sigma}_1\left(V_1 \cdot f\right)\right)\right), \tag{95}$$

ending the proof of the matrix representation. □

An immediate consequence of this Lemma is that:

$$\begin{cases} \|V_1\|_{\mathrm{op},\infty} = \max(1, \|W_1\|_{\mathrm{op},\infty}), \\ \|V_2\|_{\mathrm{op},\infty} = \max(1, \|W_2\|_{\mathrm{op},\infty}), \\ \|V_3\|_{\mathrm{op},\infty} = \|W_3\|_{\mathrm{op},\infty} + 1, & \text{if } W_s = I, \\ \|V_3\|_{\mathrm{op},\infty} \leq \|W_3\|_{\mathrm{op},\infty} + \|W_s\|_{\mathrm{op},\infty}, & \text{otherwise.} \end{cases} \tag{96}$$

### B.3  Specific structure of MobileNetV2

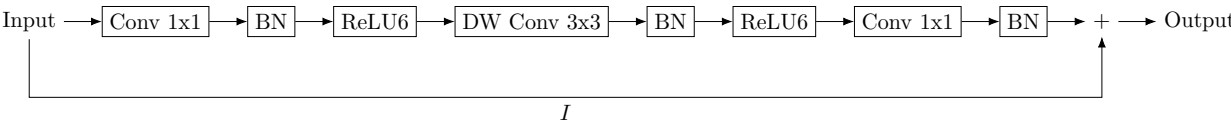

Figure 8: Structure of a Residual Block of MobileNetV2

*Remark* B.3. The only difference between Resnet-50 bottleneck block and MobileNetV2 bottleneck block is that MobileNetV2 uses inverted residuals bottleneck bock (expansion before convolution) and Depthpwise convolution (only one filter is applied for each input channel), without activation function at the end of the bloc. Thus the matrix representation can also be expressed as:

$$y = V_3 \cdot \tilde{\sigma}_2\left(V_2 \cdot \tilde{\sigma}_1\left(V_1 \cdot f\right)\right), \tag{97}$$

Where all matrices are defined in Lemma B.2.

## C  Experimental setup for the MLP case

For the Figure 5, we developed four MLPs with different depths. All of them were trained on MNIST dataset during 2 epochs, using the Adam optimizer with a learning rate of 0.001 and a batch size of 64. The MLP with depth 5 has four hidden layers with sizes [1024, 512, 256, 128]. The MLP with depth 7 has six hidden layers with sizes [1024, 512, 256, 128, 64, 32]. The MLP with depth 9 has eight hidden layers with sizes [1024, 512, 256, 128, 128, 64, 64, 32]. The MLP with depth 11 has ten hidden layers with sizes [1024, 512, 512, 256, 256, 128, 128, 64, 64, 32]. The models were quantized using uniform quantization with different bit widths: 4, 8, 16, and 24 bits only on the weights.

## D  Experimental Setup of Figure 4

**Dataset:** We used the Tiny ImageNet dataset as described in Section 5. Images were resized to $224 \times 224$ resolution. For training, standard normalization and data augmentation was applied.

**Model and training:** We replaced ResNet18, ResNet50 and MobileNetV2, final classification layer by a linear layer with 200 outputs and Batch normalization layers were removed. Models were initialized from pretrained ImageNet weights. Training was done for 80 epochs using SGD with learning rate 0.01, momentum 0.9, and weight decay $5 \cdot 10^{-4}$. Batch size was 128 for both training and validation.

**AdaRound**: applied layerwise using a calibration set of 256 samples. Each layer's rounding parameters were optimized for 15 steps using Adam with learning rate $10^{-3}$ and regularization coefficient $\lambda = 0.01$.

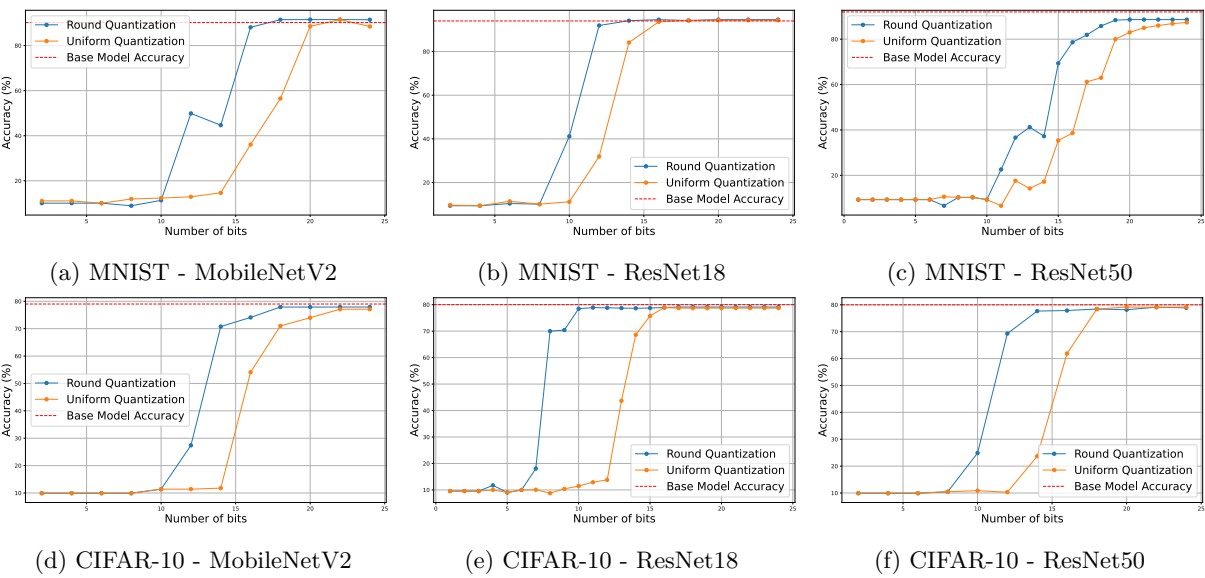

(a) MNIST - MobileNetV2     (b) MNIST - ResNet18     (c) MNIST - ResNet50

(d) CIFAR-10 - MobileNetV2     (e) CIFAR-10 - ResNet18     (f) CIFAR-10 - ResNet50

Figure 9: Graphs illustrating the effect of quantization on performance for two quantization functions (round and uniform). The results highlight how quantization reduces memory requirements while maintaining or approaching the base model's accuracy. The amount of quantization needed to reach the base precision depends on the quantization function used.

## E  Experiences on MNIST and CIFAR-10 datasets

We use two widely-used image classification datasets. The MNIST dataset: 28x28 grayscale images of handwritten digits, with 60,000 training samples and 10,000 test samples across 10 classes. The CIFAR-10 dataset: 32x32 color images in 10 different classes, with 50,000 images for training and 10,000 for testing. In Figure 9, we analyze the impact of post-training quantization on the precision of various pretrained models, across two datasets: MNIST and CIFAR-10. We clearly observe the behavior reflected by the form of the bound: depth significantly influences quantization error. For instance, in the case of ResNet50 on MNIST, uniform quantization impacts precision as the model fails to reach the baseline precision even at 24 bits. Conversely, ResNet18 achieves the baseline precision with just 12 bits on MNIST (using uniform quantization). To a lesser extent, we can also suppose that other parameters play a role. For example, despite having very similar depths, MobileNetV2 and ResNet50 exhibit noticeably different quantized performances. This is probably due to the specific architecture of MobilNetV2 that uses residual bottlenecks and Relu6 activation function ($Relu6(x) := \min(\max(0, x), 6)$ which is known to better support quantization Sandler et al. (2018). Specifically, MobileNetV2 reaches baseline precision with 20-bit quantization on MNIST.

Furthermore, the dataset influences precision as well. On CIFAR-10, ResNet50 handles quantization much better, achieving the desired precision with only 18 bits of quantization (14 bits with rounding quantization). Finally, the quantization method itself significantly affects the results: rounding quantization offers an average gain of approximately 4 bits, regardless of the dataset or model, to achieve the desired precision. This behavior is accounted for in the bound through the factor involving $\|\theta - \theta'\|_\infty$.

## F  Experimental Setup of Figure 9

**Dataset:** We used MNIST and CIFAR-10 datasets as described in Appendix E. All images were resized to $224 \times 224$ to be compatible with our three pretrained architectures. Input normalization was applied for each channel. For MNIST, which contains grayscale images, the single channel was duplicated to produce 3-channel inputs. On MNIST, only 30% of the training and test sets were used, with a batch size of 32. On CIFAR-10, the full dataset was used with a batch size of 64.

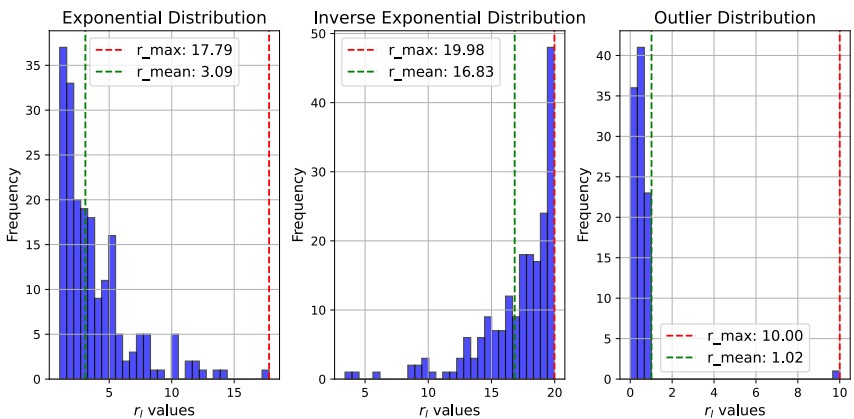

Figure 10: Comparison between the $r_{mean}$ (green) used in Theorem 4.1 and $r$ (red) used in Theorem 3.6, for three different simulated distributions, showing a smaller value compared to $r$ for each distribution.

**Models and training on MNIST:** All models were initialized with ImageNet-pretrained weights. Batch normalization layers were removed from each architecture and replaced with identity layers. For all models, the final classification layer was replaced with a new linear layer with 10 outputs. ResNet18 and ResNet50 were fine-tuned by unfreezing only the last residual block (`layer4`) along with the final linear layer (`fc`), and trained for 1 epoch. In the case of MobileNetV2, only the last convolutional block (`features[-1]`) and the classifier were unfrozen. The rest of the network remained frozen. Training was performed for 15 epochs using the Adam optimizer with a learning rate of 0.001.

**Models and training on CIFAR-10:** For CIFAR-10, the full dataset was used. Batch normalization layers were also removed before training. The ResNet18 model was trained for 5 epochs, with `layer1`, `layer4`, and `fc` layers unfrozen. The ResNet50 model was trained for 20 epochs with all layers unfrozen. MobileNetV2 was trained for 35 epochs. In this case, the convolutional blocks `features[0]`, `features[1]`, `features[17]`, `features[18]` and the classifier were the only unfrozen layers. All training process used the Adam optimizer with a learning rate of 0.001.

# G   Comparison between $r_{mean}$ and $r_{max}$

In Figure 10 we simulate several values of $r_\ell$ with different distributions. For the exponential distribution, there is variability across layers, most of $r_\ell$ values are small (less than 5) but the max is around 18. For this case we have the value of $r_{mean}$ around 3. The opposite case is the second histogram where most of $r_j$ values are large (more than 15), but even in this case, $r_{mean}$ allows to better fit to the distribution, taking account of the few small values of $r_j$. Finally, the last scenario, is the best for our bound, because for all layers except the last one, $r_j \in [0, 1]$ and the last one is 10. With this distribution we have that, $r_{mean}$ is equal to 1.02. We will see in Section 5 practical examples, and especially MobileNetV2, that is close to the exponential distribution, making us gain orders of magnitude as the approximation constant grows exponentially with respect to the depth (with parameter $r_{mean}$).

# H   Cross layer equalization as a preprocessing

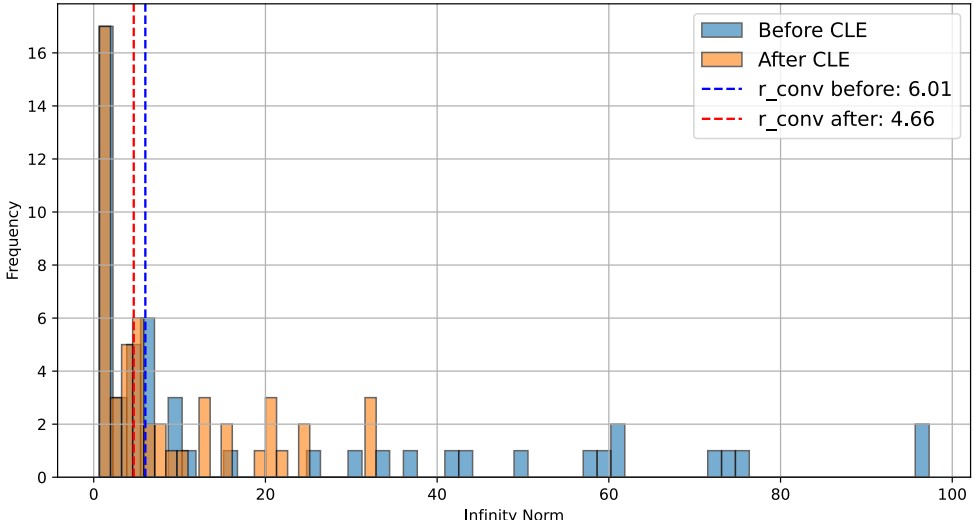

Figure 11: Comparison between weight norm distribution before and after cross-layer equalization on a 4-bits ResNet50, showing a smaller value of $r_{mean}$ after CLE.

In Figure 11 we notice that the cross layer equalization have uniformized the distribution of weight norms and therefore the value of $r_{conv}$ decreased as well. Thus our bound can be combined with preprocessing tends to reduce its pessimism, even if it is not sufficient to have a tight bound.

# I   MLPs with non-ReLu activation function

Settings of Figure 12 is exactly the same as Appendix C but we replace de ReLu activation functions, by sigmoid, implemented by Tanh, to show that our approach is robust to other activation functions.

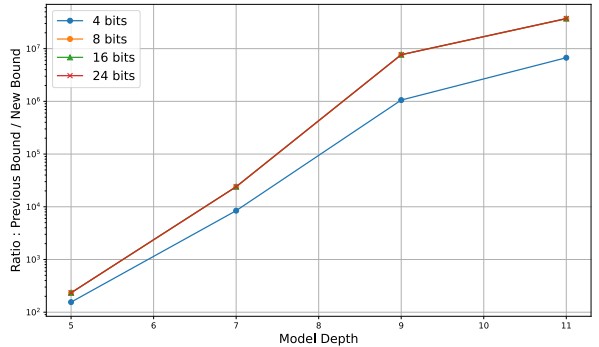

(a) Ratio in log scale between the previous bound equation 5 and our bound equation 8 as a function of model depth for different quantization bit-widths.

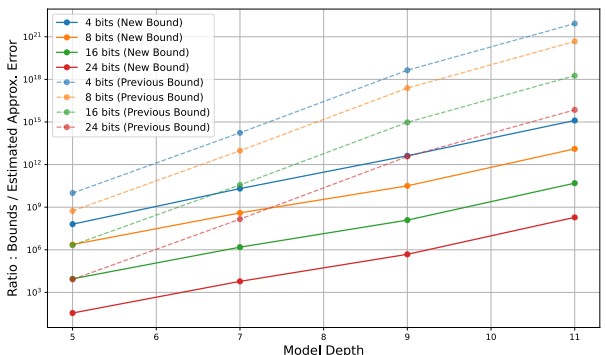

(b) Ratio in log scale between bounds equation 5, equation 8 and the estimated error approximation, as a function of model depth for different quantization bit-widths.

Figure 12: Comparison for MLPs of depths 5, 7, 9 and 11 on MNIST. (a) shows how the ratio of our bound over the previous bound grows exponentially with depth, and (b) demonstrates that our bound reduces that exponential dependence across bit-widths.

## J  Hardware

Most experiments in this paper, including all MLP evaluations and quantization analyses on MNIST and CIFAR-10, were conducted on a MacBook Air with an Apple M2 chip (8-core CPU, 8-core GPU) and 16 GB of unified memory. The experiments shown in Figure 4 were performed on an NVIDIA DGX A100 system. The DGX hardware allows us to retrain full CNN architectures on the Tiny ImageNet dataset. We emphasize that execution time and efficiency metrics are not the focus of this work. As our contribution is primarily theoretical, experiments are here only to validate and illustrate the behavior of our approximation bounds in practice.

