# OpenReview forum: "On the impact of the parametrization of deep convolutional neural networks on post-training quantization"
_TMLR — Accepted by TMLR_

### Review · Reviewer_CSVC · 2025-11-08

**Summary Of Contributions:**

The paper proposes a new theoretical approximation bound for deep (convolutional) neural networks. The authors claim that the new bound is significantly tighter than prior results (Gonon et al. 2023) across a range of architectures (ResNets, MobileNetV2). The focus is post-training quantization (PTQ): the bound is used to more accurately quantify approximation error under quantization than previous bound (Gonon et al. 2023). In addition, the authors provided experiments to validate their theoretical findings and visualize the gap between the proposed bound and the estimated approximation error.

**Audience:**

Yes

**Audience Explanation:**

Quantization is a core technique for efficient model deployment. Establishing a tighter theoretical upper bound on approximation error is essential for achieving reliable and predictable quantization.

**Broader Impact Concerns:**

Not applicable.

**Claims And Evidence:**

Yes

**Claims Explanation:**

Yes. The high-level claim of a tighter approximation bound is supported by the theoretical results and the experimental results.

**Requested Changes:**

1. Please formally define the $L_{\infty}$ norm for $\theta$, where $\theta$ is the collection of all the weight matrices before using it in Eqn (4).
2. Bound for ResNet: The main theorems (Theorems. 4.1 and 4.4) do not directly handle residual connections, and the appendix handles them only by rewriting a residual block as a wider feedforward block via adding an identity matrix (Lemma B.1 and Lemma B.2). I suggest including a corollary in the main text that explicitly presents the bound for ResNet architectures.
3. $ \left\lVert \theta - \theta^{'} \right\rVert $ is a key component of the upper bound, and its magnitude depends on the specific post-training quantization method and associated hyperparameters.I suggest that the authors include a table summarizing the upper bounds of $ \left\lVert \theta - \theta^{'} \right\rVert $ under different commonly used post-training quantization methods. This would make the implications of the theoretical results more concrete and help readers connect the theory to practical quantization techniques.
4. Please clarify whether the theoretical results apply exclusively to ReLU activations, or if they can be extended to other activation functions. If the analysis relies on specific properties of ReLU (e.g., piecewise linearity or Lipschitz continuity), it would be helpful to state this explicitly and discuss how the results might change under different activation functions.

---

> ### Author Response · Authors · 2025-11-18
>
> We thank the reviewer for the positive assessment and the very helpful suggestions. Below we address each requested change.
>
>  ## Formal definition of the \\(L\_\infty\\) norm for \\(\theta\\).
>
> We will add an explicit definition of the parameter norm used in Equation (4).
> In the general fully connected setting:
>
> $$
> \\theta = (\\tilde W\_1,\dots,\\tilde W\_L),
> $$
>
> where each \\(\\tilde W\_\ell \in \mathbb{R}^{N\_\ell \times (N\_{\ell-1}+1)}\\) includes the bias as its last column, we will define
>
> $$
> \\|\\theta\\|\_\infty := \\max\_{\ell=1,\dots,L} \\|\\tilde W\_\ell\\|\_\infty
> \quad \\text{with} \quad
> \\|\\tilde W\_\ell\\|\_\infty := \\max\_{i,j} |(\\tilde W\_\ell)\_{ij}|.
> $$
>
> Since in our analysis the biases are not quantized and are kept identical in \\(\\theta\\) and \\(\\theta'\\), the difference \\(\\|\\theta - \\theta'\\|\_\infty\\) reduces to the maximum absolute difference over all weight entries.
>
> $$
> $$
>
> ## Corollary (Approximation bound for a stack of ResNet-18 residual blocks).
>
> We propose this corollary for the specific ResNet-18 architecture:
>
> Corollary : Consider a sequence of \\(B\\) residual blocks defined in Lemma B.1, without batch normalization (as in our experimental setting).
> We assume that the activation \\(\\sigma\\) is 1-Lipschitz and satisfies \\(\\sigma(0)=0\\).
>
> Let \\(\\theta\\) and \\(\\theta'\\) denote the collections of all weight matrices
>
> $$
> \\theta = (W\_1^{(1)}, W\_2^{(1)}, \dots, W\_1^{(B)}, W\_2^{(B)}), \qquad
> \\theta' = (W\_1^{(1)\\prime}, W\_2^{(1)\\prime}, \dots, W\_1^{(B)\\prime}, W\_2^{(B)\\prime})
> $$
>
> So we have \\(L = 2B\\) layers, where the \\(\ell\\)-th layer corresponds to either \\(W\_1^{(b)}\\) or \\(W\_2^{(b)}\\) depending on \\(b\\).
> Consider additionally shortcut matrices \\(W\_s^{(b)}\\) defining skip connections for each block.
>
> $$
> r\_\ell :=
> \begin{cases}
> \\max(1,\\|W\_1^{(b)}\\|\_{\mathrm{op},\infty}), & \\text{if } \ell = 2b-1, \\\\[0.3em]
> \\|W\_2^{(b)}\\|\_{\mathrm{op},\infty} + \\|W\_s^{(b)}\\|\_{\mathrm{op},\infty}, & \\text{if } \ell = 2b.
> \end{cases}
> $$
>
> Define
>
> $$
> r\_{\mathrm{mean}}^{\mathrm{ResNet}} :=
> \\sqrt[2B-1]{\\max\_{l=1,\dots,2B} \\max\_{i=1,\dots,l-1}
> \prod\_{\substack{j=i \\\\ j\\neq l}}^{2B} r\_j }.
> $$
>
> Thus, we get the following bound over any input domain \\(\Omega \subset [-D,D]^{N\_0}\\):
>
> $$
> \\sup\_{x \in \\Omega}
> \\| R\_\\theta(\\tilde x) - R\_{\\theta'}(\\tilde x)\\|\_\infty
> \leq
> \\max(D,1)\,
> (r\_{\mathrm{mean}}^{\mathrm{ResNet}})^{2B-1}
> \sum\_{\ell=1}^{2B} p\_\ell^2 c\_{\ell-1}\,\\|\\theta - \\theta'\\|\_\infty.
> $$
> The proof can be obtained by applying Theorem 4.4 to this architecture.
> A similar corollary can be established for the ResNet-50 architecture.
>
> $$
> $$
>
> ## Table of upper bounds for \\(\\|\\theta - \\theta'\\|\_\infty\\) under different PTQ methods.
>
> For some PTQ methods, we can derive simple worst-case bounds based on the structure of the quantizer:
>
> $$
> \\sup\_w |w - Q(w)| \le \\varepsilon\_Q.
> $$
>
> with \\(Q\\) the quantization method, for example:
>
> - for uniform round-to-nearest quantization with step \\(\\eta\\), $\quad$  \\(\\varepsilon\_Q = \\eta/2\\)
> - for per-tensor uniform quantization with dynamic range \\([a,b]\\) and \\(B\\) bits,  $\quad$  \\(\\varepsilon\_Q = (b-a)/(2(2^B - 1))\\)
>
> For calibration-based methods such as AdaRound or GPTQ, the quantized weights are obtained by solving a data-dependent optimization problem.  In that case, the worst-case upper bound remains of the same form, but the actual value of  \\(\\|\\theta - \\theta'\\|\_\infty\\)  is smaller and can be measured after quantization.
>
> Furthermore, because our goal was to derive a theoretical bound that is independent of the quantization method, we intentionally kept the term  \\(\\|\\theta - \\theta'\\|\_\infty\\)  to evaluate it empirically.
>
> $$
> $$
>
> ## Applicability beyond ReLU activations.
>
> Our analysis does not rely on piece-wise linearity, it only requires that the activation \\(\\sigma\\) be 1-Lipschitz and satisfy \\(\\sigma(0)=0\\).
> Therefore, all our theorems apply to any activation with these properties, such as ReLU, leaky ReLU, \\(\\tanh\\), or softsign.
>
> We will make this assumption explicit in the main text. As mentioned in Section 5.4, we illustrate this generality in Appendix I, where we report experiments with \\(\\tanh\\), showing consistent behavior of the bound.

---

### Review · Reviewer_RXjA · 2025-11-10

**Summary Of Contributions:**

The paper introduces an analysis on quantization error under the infinity norm constraint. This means deriving tighter bound for the problem sup_x ||R_theta(x) - R_theta'(x)||_inf <= C ||\theta - \theta'||_inf where the tightness is based on the explicitation of the constant C. The improvement over prior bounds is by replacing the width parameter multiplied by number of layers by the sum of widths of all layers in the model, the operator norm of the model is also replaced by the geometric mean of the operator norms of the various matrices in the model. Equations (6)-(8) capture the contribution of the work.

In terms of experimental validation of the bounds, the authors conduct some analyses on Resnets and Mobilenets. It is observed that the new bounds do indeed improve on the prior wone from Gono et al., however the gap to the actual approximation error is still large.

**Audience:**

No

**Audience Explanation:**

As mentioned above, while the math is simple enough and convincing, in practice the proposed bounds are too loose to be useful. I wonder if this caused by the restriction to study infinity norms. Perhaps a statistical studies can result in more useful bounds on the errors with quantization. I suggest the authors to look at the relevant literature - my knowledge in that field may be a bit outdated, but at least I can point to [1] which did provide a meaningful analysis on statistical errors of the quantized models.

[1] Sakr, Charbel, Yongjune Kim, and Naresh Shanbhag. "Analytical guarantees on numerical precision of deep neural networks." International Conference on Machine Learning. PMLR, 2017.

In addition to the concern about the looseness of the bounds, I am also concerned about the empirical exploration using outdated models such as Renets and Mobilenets. Can a similar study (possibly improved by making the bounds tighter) be made using modern models such LLMs and ViTs?

**Broader Impact Concerns:**

No such concerns.

**Claims And Evidence:**

Yes

**Claims Explanation:**

It is clear that the upper bounds on the errors are indeed upper bounds since they always do results in lines "above" that of the actual approximation error. Furthermore, the theoretical derivation does not appear to have any major fallacy. Therefore, the claims made are indeed convincing. Having said that, the very large gaps to the actual approximation errors do make me wonder about how useful this analysis would be.

**Requested Changes:**

If possible, consider supplementing the analysis by a statistical approach in order to obtain tighter bounds. Open to other ideas as well, but I strongly encourage the authors to pursue tighter bounds in order to make the work useful.

In addition, please consider supplementing the empirical exploration by more contemporary models such as LLMs and ViTs.

---

> ### Author Response · Authors · 2025-11-18
>
> We thank the reviewer for the thoughtful suggestions.
>
> ## Statistical study
>
> We fully agree that complementing our analysis with a statistical study,  is an important and meaningful direction. Statistical bounds can indeed provide tighter estimates of the quantization error, which is often much smaller than the worst case deviation. However, critical safety applications such as automotive, aerospace, medical imaging, industrial control, or any setting requiring certification, legally require guarantees on the worst case error. In these contexts, a deterministic upper bound is necessary.
>
> Moreover, in the paper [1] (that we will add in the references of the revised version) they develop a probability approach that yields much more informative bounds than worst-case bounds. However, they make distributional assumptions on the quantization error of the weights and activations, and we wanted to avoid theses assumptions for our deterministic framework.
>
> We established deterministic bounds that significantly improves the existing worst case guarantees, and that remains valid for any input distribution and any quantization technique. Understanding how far deterministic approaches can lead us is, in our opinion, a necessary step to use probabilistic approaches with hypotheses as weak as possible (as is required by safety-critical applications).
>
> ## Extension to ViTs/LLM
>
> About the extension to modern architectures such as ViTs:
> as stated in the conclusion, our theoretical framework can be extended to Vision Transformers, but this would requires additional work, in particular to handle attention mechanisms and normalization layers. We chose to first refine and improve the theory in the CNN setting because for us, achieving tight and well understood bounds in this simpler setting is a necessary step before tackling more complex architectures.
>
> Besides, while the original ViT architecture does not rely on convolutional blocks, many modern vision transformer (e.g., CvT, CoAtNet, MobileViT) explicitly integrate convolutional components (convolutional stems, convolutional projections, or hybrid CNN–transformer blocks). In that sense, our analysis for CNNs can be lincked with such architectures. Extending the theoretical framework to the attention and normalization components of ViTs remains future work.
>
> As for LLM, they are out of the scope of our article focusing architectures used in safety-critical applications, but a dedicated study of the quantization of such architectures would certainly be of interest.

---

### Review · Reviewer_SkMp · 2025-11-17

**Summary Of Contributions:**

This paper tightens theoretical bounds on how much CNN outputs can change after weight quantization. The key result is a bound of the form $\sup_{x \in \Omega} \|R_\theta(x) - R_{\theta'}(x)\|_\infty \leq C\|\theta - \theta'\|_\infty$, where the constant $C$ is significantly improved over prior work (Gonon et al. 2023).

Main improvements:
- Replaces $NL^2$ dependence with $NL$ (one less factor of depth)
- Uses geometric mean of layer norms $r_{mean}$ instead of maximum $r$, which helps when weight norms vary across layers
- Exploits CNN sparsity structure (filter sizes rather than full layer widths)
- Removes the $r \geq 1$ constraint

The result is bounds that are $10^8$ tighter for ResNet18 and up to $10^{56}$ tighter for MobileNetV2. The authors validate on pretrained models with clear experimental comparisons.

Strengths: Substantial quantitative improvement, practical validation on real architectures, relevant to safety-critical applications.

Weaknesses: Bounds still far from empirical errors, requires removing BatchNorm (though this is a shared limitation with prior work), proofs are standard—the contribution is in careful constant tracking.

**Audience:**

Yes

**Audience Explanation:**

Quantization is widely used for deploying CNNs in resource-constrained and safety-critical settings (autonomous vehicles, medical imaging, robotics). While practitioners typically validate quantization empirically, theoretical bounds can inform bit-width selection and provide worst-case guarantees for safety-critical systems.

The paper also offers practical insights. E.g., MobileNetV2's weight distribution leads to much tighter bounds than ResNet50 despite similar depths. This kind of observation is useful for understanding which architectures are more quantization-friendly.

Researchers in neural network compression, quantization theory, and safety-critical ML would find this relevant.

**Broader Impact Concerns:**

None. This is theoretical work on approximation bounds with no obvious negative societal implications.

**Claims And Evidence:**

Yes

**Claims Explanation:**

The theoretical claims are clearly stated and proofs follow standard techniques. I didn't verify every step, but they appear sound and the experiments are consistent with the theory.

The experimental validation is solid:
- Table 1 shows clear parameter comparisons
- Figures 1, 3a, 3b demonstrate claimed improvements
- Bound curves track empirical error trends

The paper is generally well-written. A few areas could be clearer:
1. Section 5 mentions removing BatchNorm gives "similar accuracies" but doesn't quantify this. Practitioners need to know the actual cost.
2. Could be helpful to discuss why bounds remain ~$10^{30}\times$ larger than observed errors even after improvement, and what this means for practical use.

**Requested Changes:**

Would strengthen the work (not critical for acceptance):

1. Report accuracy with/without BatchNorm for each architecture so practitioners can assess the practical cost.

2. Even with $10^{56}\times$ improvement, bounds are far from observed errors. A paragraph on why this gap exists or future directions could be valuable.

3. Figure 2 y-axis "Frequency" could be clearer (frequency of layers?).

---

> ### Author Response · Authors · 2025-11-18
>
> We thank the reviewer for the careful reading. Below we address each requested clarification.
>
> ## Accuracy with and without BatchNorm
>
> We agree that providing quantitative information is valuable for practitioners. In the revised version, we will add the explicit information. Across all architectures, the degradation remains small (below 1% for ResNet18 and MobileNetV2, and around 1% for ResNet50).
>
> ## Why the bounds remain much larger than empirical errors
>
> Indeed the worst-case bound remains significantly above the empirical approximation error. As we said in the paper (p8-9), one reason is that such bounds are derived under theoretical worst-case scenarios, which rarely occur in practice. Another reason is the generality of the theorem itself, which leads to a very conservative estimate of the output error.
>
> Besides, layer-wise analysis relies on inequalities such as:
>
> $$
> \|A x\|\_\infty \le \|A\|\_{\mathrm{op},\infty}\,\|x\|\_\infty
> $$
>
> which become conservative when the activations are not aligned with the worst-case directions of the matrices. This type of misalignment can explain the gaps between worst-case and typical behavior. We will emphase these mechanisms in the conclusion.
>
> Finally, the purpose of our analysis is to provide formal worst-case guarantees that apply to all inputs and all quantization schemes which play an important role in safety-critical applications.
>
> ## Clarification of Figure 2
>
> We thank the reviewer for pointing this out. In the revised version, we will clarify that the y axis represents the frequency of the layers.

---

### Decision · Action_Editor_4SkA · 2026-01-14

**Recommendation:** Accept with minor revision

**Audience:**

Yes

**Audience Explanation:**

This work is of interest to the ML community, in particular, those interested in compression, quantization, and safety-critical ML.

**Claims And Evidence:**

Yes

**Claims Explanation:**

The authors have studied how much CNN activation outputs (layer-wise) change due to post-training quantization. They look at variations of CNN outputs under norm infinity over specific domains for parameters and input vectors. The authors have shown that their analyses result in tiger bounds on the quantization error compared to previous work (Gonon et al. 2023).
The key technical improvement is by replacing the width parameter multiplied by number of layers by the sum of widths of all layers in the model and replacing the operator norm of the model by the geometric mean of the operator norms of the various matrices in the model. The authors have further provided experiments across Resnet and Mobilenet architectures.

The reviewers have identified key strengths in this paper including “Substantial quantitative improvement, practical validation on real architectures, relevant to safety-critical applications” (Reviewer SkMp) and “the authors provided experiments to validate their theoretical findings and visualize the gap between the proposed bound and the estimated approximation error” (Reviewer CSVC). I would recommend the paper to be accepted after the authors address reviewers' comments in the camera-ready version.

This work is of interest to the ML community, in particular, those interested in compression,  quantization, and safety-critical ML.